# TetraGT: Tetrahedral Geometry-Driven Explicit Token Interactions with Graph Transformer for Molecular Representation Learning

**Jinjia Feng**[1]    **Zhewei Wei**[1*]  **Taifeng Wang**[2]    **Zongyang Qiu**[2]

[1]Gaoling School of Artificial Intelligence, Renmin University of China, Beijing, China
[2]BioMap Research, Beijing, China

`{jinjia_feng, zhewei}@ruc.edu.cn`,`tfwangster@gmail.com`,
`qiuzongyang23@outlook.com`

## Abstract

Molecular representations that fully capture geometric parameters such as bond angles and torsion angles are crucial for accurately predicting important molecular properties including enzyme catalytic activity, drug bioactivity, and molecular spectral characteristics, as demonstrated by extensive studies. However, current molecular graph representation learning approaches represent molecular geometric parameters only indirectly through combinations of atoms and bonds, neglecting the spatial relationships and interactions between these higher-order geometric structures. In this paper, we propose **TetraGT** (**Tetra**hedral **G**eometry-Driven Explicit **T**oken Interactions with Graph Transformer), a novel architecture that directly models molecular geometric parameters. Based on the spatial solid geometry theory of face angle and dihedral angle inequality, TetraGT explicitly represents bond angles and torsion angles as structured tokens for the first time, directly reflecting their intrinsic role in determining the molecular conformational stability and properties. Through our designed spatial tetrahedral attention mechanism, TetraGT achieves highly selective direct communication between structural tokens. Experimental results demonstrate that TetraGT achieves superior performance on the PCQM4Mv2 and OC20 IS2RE benchmarks. We also apply our pre-trained TetraGT model to downstream tasks including QM9, PDBBind, Peptides and LIT-PCBA, demonstrating that TetraGT delivers excellent results in transfer learning scenarios and shows scalability with increasing molecular size. Our code is available at `https://github.com/xkxxfyf/TetraGT`.

## 1 Introduction

Accurate prediction of molecular properties is essential for advancements in drug discovery and biochemical research. Properties such as enzyme catalytic activity (Lopes et al., 2009), drug bioactivity (Kessler, 1982), molecular spectral characteristics (Wald, 1968), and reaction stereoselectivity (Hashimoto & Maruoka, 2015) are intrinsically determined by the three-dimensional structural arrangements of molecules. Studies have consistently shown that molecular geometric parameters—particularly bond angles and torsion angles—critically influence these properties by determining molecular conformational stability and reactivity (Lovering et al., 2009).

Following the success of Transformer (Vaswani, 2017) in various domains, Graph Transformers (GTs) (Ying et al., 2021; Feng et al., 2022) have emerged as powerful approaches to learning molecular representation. Recent GTs (Zhou et al., 2023; Stärk et al., 2022; Hussain et al., 2022), which incorporate 3D structural information, represent molecules primarily through node tokens (atoms) and sometimes edge tokens (bonds), relying on global attention mechanisms to facilitate information

---

*Zhewei Wei is the corresponding author.

exchange across the molecular graph. Inspired by the remarkable success of AlphaFold (Jumper et al., 2021) in protein structure prediction, methods such as UniMol+ (Lu et al., 2023) and TGT (Hussain et al., 2024) have introduced triangle inequality-constrained interatomic distance prediction. Such advancements have demonstrated that the incorporation of geometric constraints can significantly enhance the prediction performance of molecular properties.

Despite these improvements, current approaches encounter three primary challenges in representing molecular geometry comprehensively. **(1) Lack of Local Chirality:** Chirality describes the "handedness" of molecular spatial arrangements. Molecules with different local chirality may yield similar or even identical distance matrices, creating ambiguity in molecular representation. **(2) Implicit Modeling of Geometric Structures:** Recent models such as QuinNet (Wang et al., 2024c) and ViSNet (Wang et al., 2024b) that introduce four- or five-atom interactions to enhance expressiveness, still encode higher-order geometric information only implicitly through combinatorial operations between atom-level tokens, causing deviations in geometric parameters to propagate and accumulate through indirect representations. **(3) Structural Interdependency Neglect:** Existing approaches fail to account for the interdependent relationships between molecular geometric parameters, relations that collectively determine the overall molecular conformation and subsequently its functional properties.

To address these challenges, we propose TetraGT, a novel Graph Transformer architecture that explicitly models geometric parameters as structured tokens. TetraGT directly represents bond angles and torsion angles as structural tokens in the model architecture, reducing the accumulation of prediction errors. TetraGT introduces a spatial tetrahedral attention mechanism that promotes effective communication between geometric-parameter tokens while ensuring the satisfaction of tetrahedral geometric constraints, enabling the generation of globally consistent and physically realistic molecular conformations. In addition, we introduce a directed cycle angle loss that enables stable prediction of angles over the full range $(0, 2\pi)$ and leverages directionality for explicit local chirality discrimination. Furthermore, based on a hierarchical virtual node aggregation architecture, our method effectively alleviates information bottlenecks caused by excessive compression of information from different orders. By learning fundamental geometric principles rather than memorizing specific conformational patterns, TetraGT improves the model's generalization ability across diverse molecular structures and can accurately predict geometries from scratch, **without requiring initial estimates of 3D coordinates**. Our main contributions are summarized as follows:

- To the best of our knowledge, we are the first to explicitly model molecular geometric parameters (bond angles and torsion angles) as structured tokens rather than indirectly deriving them from pairwise atomic relationships. This direct representation significantly reduces accumulated errors and enables more accurate capture of molecular conformational characteristics.

- We introduce three synergistic architectural innovations—spatial tetrahedral attention mechanism, directed cycle angle loss, and hierarchical virtual node aggregation architecture—that collectively enhance TetraGT's expressive power by enforcing geometric consistency, enabling chirality discrimination, and capturing multi-scale structural information.

- TetraGT achieves superior performance across multiple benchmarks, establishing new state-of-the-art results on quantum chemistry datasets (PCQM4Mv2 and OC20 IS2RE) and demonstrating remarkable effectiveness in diverse transfer learning scenarios including molecular property prediction (QM9 and Peptides), binding affinity prediction (PDBBind), and drug discovery (LIT-PCBA).

## 2 RELATED WORK

### 2.1 ANGLE PREDICTION IN MOLECULAR CONFORMATION OPTIMIZATION

The incorporation of angular constraints, including bond angles and torsion angles, in molecular conformations has been progressively applied in recent works. Early methods (Ganea et al., 2021; Rai et al., 2022) introduced torsion angle constraints in three-dimensional conformation generation. Building on these foundations, several diffusion-based and autoregressive approaches have emerged (Jing et al., 2022; Zhang et al., 2023). For example, DiffPack (Zhang et al., 2024) learned the joint distribution of side-chain torsion angles through diffusion and denoising and AUTODIFF (Li et al., 2024a) designed conformational motifs as a molecular assembly strategy to mitigate issues with skewed angles. Other methods attempt to capture angular information more explicitly through equivariance and positional encoding that incorporates angular information, such as LEFTNet (Du

et al., 2024), SaVeNet (Aykent & Xia, 2024) and Geoformer (Wang et al., 2024a). However, these methods still predict angles implicitly from atomic coordinates or encode them as auxiliary features rather than modeling them as explicit structural entities, leading to accumulated errors as deviations compound through indirect representations. More critically, none of these works address the fundamental challenge of local chirality discrimination—molecules with different chirality can produce similar angle and distance distributions, creating inherent ambiguity (Lin et al., 2026). **Our approach fundamentally differs by treating geometric parameters as structured tokens**, with bond angles and torsion angles represented as first-class entities. We introduce a **directed cycle angle loss** that predicts angles in the full range of $(0, 2\pi)$, incorporating directionality to explicitly capture local chirality, making TetraGT the first work to achieve chirality-aware molecular representation through angular modeling.

## 2.2 PREDICTIVE MOLECULAR STRUCTURAL PRE-TRAINING

AlphaFold (Jumper et al., 2021) employs a Transformer architecture for predictive structural pre-training on protein datasets. In small molecule structural pre-training, hybrid approaches such as GraphTrans (Wu et al., 2021), GSA (Rashedi et al., 2009), GROVER (Rong et al., 2020), and GPS (Rampášek et al., 2022a) combine Transformers with Graph Neural Networks for enhanced expressiveness. Pure GT models directly take structural elements as token inputs, where Graphormer (Ying et al., 2021; Shi et al., 2022) and EGT (Hussain et al., 2022) are the two most representative approaches. Graphormer-type models including Unimol (Zhou et al., 2023), GEM-2 (Liu et al., 2022a), and Transformer-M (Luo et al., 2022) primarily use atoms as tokens, implicitly encoding bonds and spatial structures through positional encoding and attention bias. In contrast, EGT-based models treat edge embeddings as tokens with global attention for node-edge information exchange. Recent advances have incorporated triangular inequality constraints, as seen in GPS++ (Masters et al., 2022), Unimol+ (Lu et al., 2023), and TGT (Hussain et al., 2024), improving geometric consistency at the edge level. Despite these advances, current methods remain limited to the interactions between node-level and edge-level tokens, failing to capture **structural interdependencies** among higher-order geometric structures (bond angles and torsion angles). This results in isolated modeling of bond angles and torsion angles, significantly compromising the physical plausibility of conformations. Building upon explicit modeling of angles as tokens, TetraGT introduces a **spatial tetrahedral attention mechanism** that ensures these parameters collectively satisfy inequalities derived from tetrahedral geometry, enabling effective communication between interdependent structures—extending triangle inequality principles from edges to higher-order elements. Meanwhile, our **hierarchical virtual node architecture** aggregates information across different structural orders, allowing the model to learn multi-scale interdependencies.

## 3 METHOD

### 3.1 TETRAGT ARCHITECTURE

**The Definition and Preliminaries**

The TetraGT model utilizes atomic features ($\boldsymbol{X} \in \mathbb{R}^{n \times d_x}$, where $n$ is the number of atoms and $d_x$ is the atom feature dimension), edge features ($\boldsymbol{E} \in \mathbb{R}^{n \times n \times d_e}$, where $d_e$ is the edge feature dimension), and 3D conformational information including the complete distance matrix ($\boldsymbol{D} \in \mathbb{R}^{n \times n}$), all bond angles ($\boldsymbol{B} \in \mathbb{R}^{n_b}$, $n_b$ is the number of bond angles), and torsion angles ($\boldsymbol{T} \in \mathbb{R}^{n_t}$, $n_t$ is the number of torsion angles) within the molecule to predict molecular properties $y$ and update 3D conformational information using learnable parameters $\theta$. The model has $L$ blocks, where the $l$-th block outputs node embeddings $h^{(l)}$, edge embeddings $e^{(l)}$, bond angle embeddings $b^{(l)}$, and dihedral angle embeddings $t^{(l)}$. To ensure the physical validity of predicted molecular geometries, we consider the tetrahedron as a fundamental local structure that satisfies basic geometric constraints governing the spatial arrangement of atoms. Here, "tetrahedra" refers to the geometric 3D simplex formed by any four non-coplanar atoms, rather than a chemical tetrahedral ($sp^3$) center.

Specifically, such tetrahedra must satisfy the following angle constraints:

**Lemma 1** (Tetrahedral Angle Constraints). *Let $\{i, j, k, l\}$ be four non-coplanar points in $\mathbb{R}^3$ forming a (non-degenerate) tetrahedron. In particular, all face angles and dihedral angles introduced below*

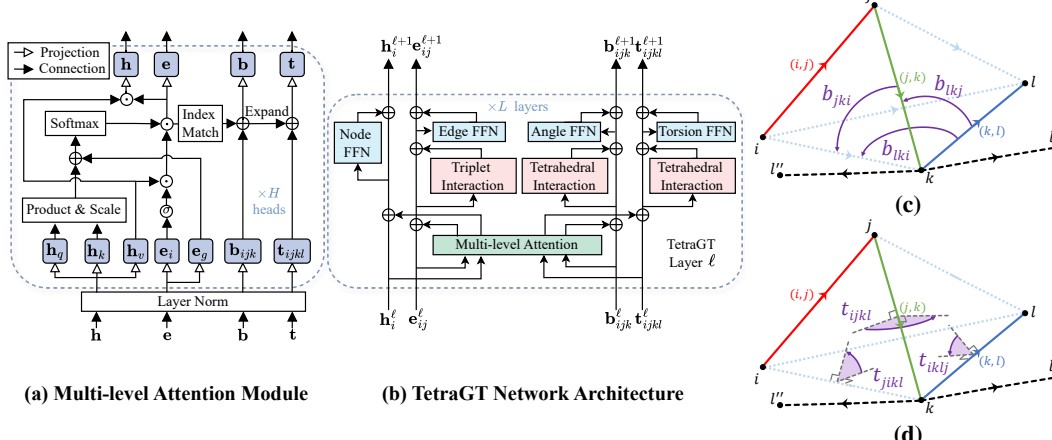

Figure 1: (a) Multi-level Attention Module. Index Match denotes the selection of corresponding edge embeddings based on the indices of nodes. (b) TetraGT Network Architecture. (c) Tetrahedral interaction for bond angles. (d) Tetrahedral interaction for torsion angles.

*lie in $(0, 2\pi)$. Let $b_{jki}, b_{lki}, b_{lkj}$ denote the face angles at vertex $k$ in faces $ijk, ikl, jkl$ respectively, and let $t_{ijkl}, t_{jikl}, t_{iklj}$ denote the dihedral angles between faces $ijk$ and $ikl$, faces $ijk$ and $jkl$, and faces $ikl$ and $jkl$ respectively. Then:*

*(a) Face Angles: For any permutation $\{\theta_1, \theta_2, \theta_3\} = \{b_{jki}, b_{lki}, b_{lkj}\}$, we have $\theta_1 + \theta_2 > \theta_3$ and $\theta_1 + \theta_2 + \theta_3 < 2\pi$.*

*(b) Dihedral Angles: For any permutation $\{\phi_1, \phi_2, \phi_3\} = \{t_{ijkl}, t_{jikl}, t_{iklj}\}$, we have $\phi_1 + \phi_2 > \phi_3$.*

*(c) Relationship between Face and Dihedral Angles at vertex $k$:*

$$\cos(t_{ijkl}) = \frac{\cos(b_{jki}) - \cos(b_{lki})\cos(b_{lkj})}{\sin(b_{lki})\sin(b_{lkj})} \tag{1}$$

*with similar formulas for $t_{jikl}$ and $t_{iklj}$ by cyclic permutation of the indices. The non-coplanarity assumption ensures that the denominators in the above expressions are non-zero.*

**The Initialization of Substructures.** Atom representations are composed of the atom's inherent properties, while edge representations are formed by chemical bond properties, the types of atoms at both ends, and bond length. As illustrated in Figure 1(c) and (d), TetraGT comprehensively models neighboring face angles (such as $b_{lkj}$, $b_{l'kj}$, and $b_{l''kj}$) between atom triplets and dihedral angles ($t_{ijkl}$, $t_{ikl'j}$, etc.) between atom quadruplets. This approach enables face angles like $b_{lkj}$, $b_{l'kj}$, and $b_{l''kj}$ to aggregate information directly without being bottlenecked by nodes $k$ and $j$, with the same principle applying to dihedral angles. Inspired by Lemma 1, we initialize dihedral angle tokens using atom and edge representations along with face angle information. This approach naturally incorporates tetrahedral geometric constraints into our model.

**Multi-level Attention Module.** In each layer, the Multi-level Attention mechanism leverages output representations from the previous layer to facilitate continued interaction and refinement of these geometric features. First, we compute the node and edge embeddings through the attention mechanism shown in Figure 1(a). Subsequently, the bond angle embedding is obtained by using the indices of the two edges forming the angle to locate the corresponding positions and summing the embeddings. Similarly, for dihedral angle updates, we use the indices of three consecutive edges that form the torsion angle to locate and sum the corresponding torsion angle embeddings. This approach allows for a hierarchical update of representations of different structural levels in the graph, progressing from atoms to chemical bonds, then to bond angles, and finally to torsion angles. The updates of atom and edge representations in the Multi-level Attention Module are as follows:

$$h^{(l)} = \text{softmax}\left(e^{(l)}\right)\sigma(e^{(l-1)}W_G^{(l,e)})h^{(l-1)}W_V^{(l,h)}, \quad e^{(l)} = h^{(l-1)}W_Q^{(l,h)}\left(h^{(l-1)}W_K^{(l,h)}\right)^T/\sqrt{d_h} + e^{(l-1)}W_E^{(l,e)}. \tag{2}$$

where $d_h$ is the head dimension, $\boldsymbol{W}_Q^{(l,h)}, \boldsymbol{W}_K^{(l,h)}, \boldsymbol{W}_V^{(l,h)} \in \mathbb{R}^{d_a \times d_h}, \boldsymbol{W}_E^{(l,e)}, \boldsymbol{W}_G^{(l,e)} \in \mathbb{R}^{d_p \times d_h}$. The representation of bond angles and torsion angles is achieved by adding the corresponding edge

representations to their respective indices:

$$b_{ijk}^{(l)} = \sum (ab) \in \{(ij), (jk)\} e_{ab}^{(l)} + b_{ijk}^{(l-1)} W_B^{(l,b)}. \tag{3}$$

$$t_{ijkl}^{(l)} = \sum (ab) \in \{(ij), (jk), (kl)\} e_{ab}^{(l)} + t_{ijkl}^{(l-1)} W_T^{(l,b)}. \tag{4}$$

where $\boldsymbol{W}_B^{(l,h)} \in \mathbb{R}^{d_b \times d_h}, \boldsymbol{W}_T^{(l,h)} \in \mathbb{R}^{d_t \times d_h}$. Both bond angles and torsion angles utilize the edge representations from the current layer for aggregation, allowing for an efficient use of atomic and edge representations from the previous layer.

**Tetrahedral Interaction Module.** We carefully designed this approach for two primary reasons. First, naively modeling all possible triplet and quadruplet interactions has unacceptable computational costs, especially when the number of atoms in molecules scales. Second, arbitrary substructures often lack physical significance in molecular modeling. TetraGT leverages tetrahedral geometric theory to selectively model meaningful higher-order interactions in tetrahedral structures. To further improve efficiency, we introduce a local sampling strategy that restricts attention to the $w$ nearest neighbors, reducing theoretical complexity from $O(N^3)$ to $O(wN^2)$.

For a central bond angle $(i, j, k)$, the facial interaction with neighboring bond angles sharing vertex $k$ is computed as follows:

$$\mathbf{o}_{jki}^{f} = \sum_{l \in \mathcal{N}_w(j)} a_{ijkl}^{f} \mathbf{v}_{lkj}^{f}, \quad a_{ijkl}^{f} = \text{softmax}_l \left( \frac{\mathbf{q}^{f}(b_{jki}) \cdot \mathbf{p}^{f}(t_{lkj})}{\sqrt{d}} + \mathbf{b}^{f}(b_{lki}) \right) \sigma(\mathbf{g}^{f}(b_{lki})) \tag{5}$$

where $\mathcal{N}_w(j)$ denotes the set of $w$ nearest neighbors of $j$ at vertex $k$, the value vector $\mathbf{v}_{lkj}^{f}$ is derived from a learnable projection of the bond angle embedding $\mathbf{b}_{lkj}$, and $a_{ijkl}^{f}$ is the attention weight assigned to the bond angle $(l, k, j)$ by the bond angle $(i, j, k)$. As shown in Figure 1(c), design enables efficient and direct information exchange between neighboring face angles like $b_{jki}$, $b_{lki}$, and $b_{lkj}$ at vertex $k$, while maintaining computational tractability through strategic local sampling. Meanwhile, the query vector $\mathbf{q}^{f}$ and key vector $\mathbf{p}^{f}$ are derived from the bond angle embeddings. The bias term $\mathbf{b}^{f}$ and gating term $\mathbf{g}^{f}$ are scalars derived from the bond angle embedding $b_{lki}$, incorporating geometric constraints from Lemma 1(a) to facilitate physically valid interactions.

Similar to face angles, for a torsion angle $(i, j, k, l)$, the dihedral interaction with neighboring torsion angles is computed as follows:

$$\mathbf{o}_{ijkl}^{d} = \sum_{l \in \mathcal{N}_w} a_{ijkl}^{d} \mathbf{v}_{iklj}^{d}, \quad a_{ijkl}^{d} = \text{softmax}_l \left( \frac{\mathbf{q}^{d}(t_{ijkl}) \cdot \mathbf{p}^{d}(t_{iklj})}{\sqrt{d}} + \mathbf{b}^{d}(t_{jikl}) \right) \sigma(\mathbf{g}^{d}(t_{jikl})) \tag{6}$$

where $\mathcal{N}_w$ represents the $w$ nearest neighboring torsion angles, and the value vector $\mathbf{v}_{iklj}^{d}$ is derived from the torsion angle embedding. As illustrated in Figure 1(d), this mechanism enables efficient direct information exchange between dihedral angles sharing common faces. The fixed atoms (e.g., $i, j, k$ in Figure 1(d)) naturally form the base bond angle, ensuring that interacting dihedral angles are not composed of completely disconnected atoms. This carefully designed local sampling strategy preserves structural validity while effectively managing the computational complexity inherent to higher-order interactions (detailed computational analysis in Appendix E). Furthermore, the bias and gating terms incorporate constraints from Lemma 1(b), informed by the face-dihedral relationship established in Lemma 1(c), to maintain geometrically consistent representations.

Following these specialized attention mechanisms, the representations are updated as:

$$\mathbf{b}^{(l)} = \mathbf{b}^{(l-1)} + \text{FFN}(\mathbf{o}_{jki}^{f}), \quad \mathbf{t}^{(l)} = \mathbf{t}^{(l-1)} + \text{FFN}(\mathbf{o}_{ijkl}^{d}) \tag{7}$$

**Directed Cycle Angle Loss (DCA loss).** TetraGT extends molecular geometry prediction to include bond and torsion angles. When chirality changes, at least one angle shifts from $\sigma$ to $2\pi - \sigma$ in a fixed reference direction. Distance-only prediction methods struggle with this distinction, as both angular values satisfy identical distance matrices. This challenge becomes particularly acute at molecular terminals, where chirality-induced distance variations become nearly imperceptible. Previous approaches often restricted angles to the 0-$\pi$ range, failing to capture chiral variations. Recognizing the cyclic nature of angle prediction, TetraGT implements a directed circular binning loss:

$$L_{DCA} = \min \left( -\sum_{i=1}^{N} q_i \log(p_i), -\sum_{i=1}^{N} q_i \log(p_{(i+1) \mod N}) \right). \tag{8}$$

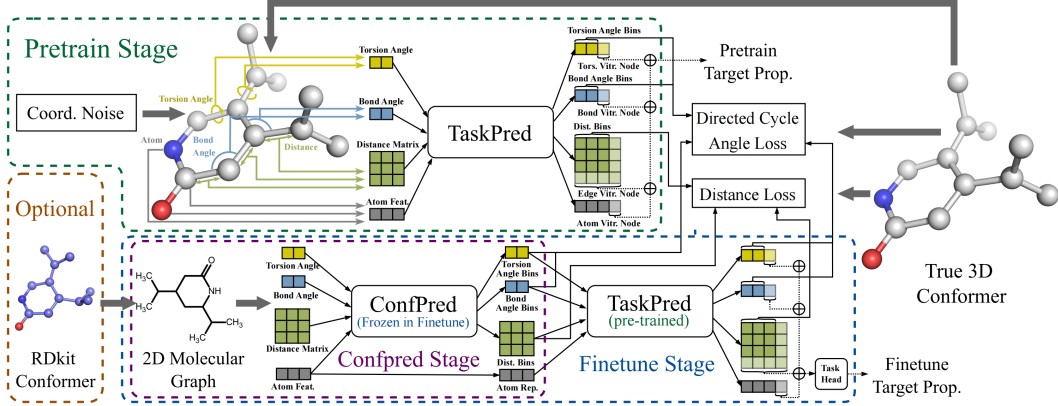

Figure 2: The three stages of TetraGT training.

By extending the angle range to $(0, 2\pi)$ with counterclockwise as the primary direction, this approach accommodates all chirality scenarios and appropriately handles boundary cases, avoiding excessive penalties for angles that are conceptually close but numerically distant (e.g., 359° vs 1°).

**Hierarchical Virtual Node.** Recent studies (Li et al., 2024b; Xing et al., 2024) show that virtual nodes in graph data significantly reduce information bottlenecks, but previous molecular property prediction approaches either compressed all atomic information (losing critical structural details) or used only atomic-level virtual nodes (inadequately representing 3D interactions). To address these limitations, TetraGT implements a novel hierarchical virtual nodes architecture. Each substructure type (atoms, edges, bond angles, torsion angles) has a dedicated virtual node that interacts with tokens of its own type through appropriate mechanisms: FFN for atoms, triplet interaction for edges, and tetrahedral interaction for bond and torsion angles. For property prediction, a molecule-level virtual node connects to these four substructure virtual nodes as the final output representation.

## 3.2 MODEL TRAINING

The training procedure of TetraGT includes three stages for the molecular property prediction task. First, in the conformation prediction stage, a conformation predictor is trained to predict accurate molecular conformations from 2D molecular graphs. During the pretraining stage, a task predictor is employed to predict molecular properties from the pre-training dataset. This predictor also receives noisy conformational structures as input and denoise conformational structures. In the fine-tuning stage, the frozen, pre-trained conformation predictor and task predictor are fine-tuned on downstream datasets.

**Conformer Prediction Stage.** We train the TetraGT conformation predictor to predict all pairwise interatomic distances, bond angles, and torsion angles within a molecule. The conformation predictor takes a 2D molecular graphs as input (Optionally, an initial distance estimate, e.g., from RDKit coordinates) and outputs all pairwise interatomic distances, bond angles, and torsion angles. Angles are invariant to translation and rotation. Inspired by TGT, we predict binned angles instead of continuous values, as torsion angle structures are typically less stable than chemical bonds and more susceptible to rapid changes due to molecular energy fluctuations. The TetraGT employs cross-entropy loss for pairwise atomic distances and the Directed Cycle Angle Loss for angles.

**Pre-training Stage.** In the pre-training phase, TetraGT trains the TetraGT task predictor on noisy ground truth 3D conformations. This approach ensures that the task predictor is robust to noise in both input distances and angles, enabling it to adapt to approximate conformations output by the conformation predictor, which still contain noise. We maintain predictions for pairwise interatomic distances, bond angles, and torsion angles. We use distance prediction loss and angle prediction loss as auxiliary tasks that encourage different order substructure representations to denoise the 3D structure, combined with the primary tasks from the pre-training dataset, to jointly train TetraGT's task predictor in a multi-task learning framework. Furthermore, TetraGT employs hierarchical substructure virtual nodes for joint prediction in molecular property prediction, facilitating the association between substructures and molecular properties.

Table 1: LIT-PCBA results. The top **1**[st] and 2[nd] results are highlighted.

| Model | Avg. Test ROC-AUC↑ (%) |
|---|---|
| NaiveBayes (Webb et al., 2010) | 73.0 |
| SVM (Hearst et al., 1998) | 73.4 |
| RandomForest (Breiman, 2001) | 62.0 |
| XGBoost (Chen & Guestrin, 2016) | 72.6 |
| GCN (Kipf & Welling, 2016) | 72.3 |
| GAT (Velickovic et al., 2017) | 75.2 |
| FP-GNN (Cai et al., 2022) | 75.9 |
| EGT (Hussain et al., 2022) | 78.9 |
| GEM (Fang et al., 2022) | 78.4 |
| GEM-2 (Liu et al., 2022a) | 81.5 |
| EGT+RDKit (Hussain et al., 2024) | 81.2 |
| TGT (Hussain et al., 2024) | 81.5 |
| TetraGT | **82.4** |

Table 2: Results on PCQM4Mv2 valid set.

| Model | # param. | MAE (meV)↓ |
|---|---|---|
| MLP-Fingerprint (Hu et al., 2022) | 16.1M | 173.5 |
| GCN (Kipf & Welling, 2016) | 2.0M | 137.9 |
| GIN (Xu et al., 2018) | 3.8M | 119.5 |
| GINEv2 (Brossard et al., 2020) | 13.2M | 116.7 |
| GIN-VN (Xu et al., 2018; Gilmer et al., 2017) | 6.7M | 108.3 |
| DeeperGCN-VN (Li et al., 2020) | 25.5M | 102.1 |
| TokenGT (Kim et al., 2022) | 48.5M | 91.0 |
| EGT (Hussain et al., 2022) | 89.3M | 86.9 |
| GRPE (Park et al.) | 46.2M | 86.7 |
| Graphormer (Ying et al., 2021; Shi et al., 2022) | 47.1M | 86.4 |
| GraphGPS (Liu et al.) | 13.8M | 85.2 |
| GEM-2(+RDKit) (Liu et al., 2022a) | 32.1M | 79.3 |
| GPS++ (Masters et al., 2022) | 44.3M | 78.1 |
| Transformer-M (Luo et al., 2022) | 69M | 77.2 |
| Uni-Mol+(+RDKit) (Lu et al., 2023) | 77M | 69.3 |
| TGT(+RDKit) (Hussain et al., 2024) | 203M | 67.1 |
| TetraGT-6 layer | 60M | 69.3 |
| TetraGT-12 layer | 127M | 68.1 |
| TetraGT-24 layer | 215M | 67.1 |
| TetraGT-24 layer(+RDKit) | 215M | **65.9** |

Table 3: Performance on OC20 IS2RE validation set.

| Model | Energy MAE (meV)↓ | | | | | EwT (%)↑ | | | | |
|---|---|---|---|---|---|---|---|---|---|---|
| | ID | OOD Ads. | OOD Cat. | OOD Both | AVG. | ID | OOD Ads. | OOD Cat. | OOD Both | AVG. |
| SchNet (Schütt et al., 2017) | 646.5 | 707.4 | 647.5 | 662.6 | 666.0 | 2.96 | 2.22 | 3.03 | 2.38 | 2.65 |
| DimeNet++ (Gasteiger et al., 2020) | 563.6 | 712.7 | 561.2 | 649.2 | 621.7 | 4.25 | 2.48 | 4.40 | 2.56 | 3.42 |
| GemNet-T (Gasteiger et al., 2021) | 556.1 | 734.2 | 565.9 | 696.4 | 638.2 | 4.51 | 2.24 | 4.37 | 2.38 | 3.38 |
| SphereNet (Liu et al., 2022b) | 563.2 | 668.2 | 559.0 | 619.0 | 602.4 | 4.56 | 2.70 | 4.59 | 2.70 | 3.64 |
| GNS (Godwin et al., b) | 540.0 | 650.0 | 550.0 | 590.0 | 582.5 | - | - | - | - | - |
| GNS+NN (Godwin et al., b) | 470.0 | 510.0 | 480.0 | 460.0 | 480.0 | - | - | - | - | - |
| Graphormer-3D (Shi et al., 2022) | 432.9 | 585.0 | 444.1 | 529.9 | 498.0 | - | - | - | - | - |
| EquiFormer (Liao & Smidt) | 422.2 | 542.0 | 423.1 | 475.4 | 465.7 | 7.23 | 3.77 | 7.13 | 4.10 | 5.56 |
| EquiFormer+NN (Liao & Smidt) | 415.6 | 497.6 | 416.5 | 434.4 | 441.0 | 7.47 | 4.64 | 7.19 | 4.84 | 6.04 |
| DRFormer (Wang et al., 2023) | 418.7 | 486.3 | 432.1 | 433.2 | 442.5 | 8.39 | 5.42 | 8.12 | 5.44 | 6.84 |
| Uni-Mol+ (Lu et al., 2023) | 379.5 | 452.6 | 401.1 | 402.1 | 408.8 | 11.1 | 6.71 | 9.90 | 6.68 | 8.61 |
| TGT (Hussain et al., 2024) | 381.3 | 445.4 | 391.7 | 393.6 | 403.0 | 11.1 | 6.87 | 10.47 | 6.80 | 8.82 |
| TetraGT | **375.3** | **440.1** | **382.8** | **392.7** | **397.7** | **11.9** | **7.28** | **11.69** | **6.90** | **9.14** |

**Fine-tune Stage.** In the fine-tuning phase, TetraGT employs a frozen, pre-trained conformation predictor to efficiently generate high-precision 3D structural features from the 2D molecule graph. During this process, the conformation predictor specifically operates in stochastic mode with active dropout (Hussain et al., 2024). Subsequently, all the predicted bond angles, torsion angles, and interatomic pair distances serve as input to the task predictor. The fine-tuning process simultaneously combines the primary objective of the downstream dataset's task with auxiliary optimization functions for distance and angle prediction. Specifically, we utilize the model-generated interatomic pair distance matrix, bond angles, and torsion angles as input, requiring the model to predict the same substructures in the ground truth conformations, as well as the target objectives of the current dataset.

## 4 EXPERIMENTS

The experimental section aims to validate the effectiveness of our proposed model and methods in addressing existing challenges. We first demonstrate the performance and scalability of TetraGT on large-scale quantum chemistry datasets, PCQM4Mv2 (Hu et al., 2022) and OC20 IS2RE (Chanussot et al., 2021). We then evaluate the transfer learning capabilities of the TetraGT model in both the conformer prediction and pre-training stages. We conduct ablation studies on the key components and aggregated angle representation of TetraGT, and analyze its efficiency and scalability across different molecular sizes. We also conducted quantitative analysis and visualization of conformer chirality prediction and conformer prediction accuracy, with details provided in Appendix C and D. Full experimental details and configurations can be found in Appendix F.

Table 4: Results (MAE($\downarrow$)) on the QM9 dataset.

| Method | $\mu$ | $\alpha$ | $\epsilon_H$ | $\epsilon_L$ | $\Delta\epsilon$ | ZPVE | $C_v$ | $U_0$ | $U$ | $H$ | $G$ | $R^2$ |
|---|---|---|---|---|---|---|---|---|---|---|---|---|
| GraphMVP (Liu et al.) | 0.031 | 0.070 | 28.5 | 26.3 | 46.9 | 1.63 | 0.033 | - | - | - | - | - |
| GEM (Fang et al., 2022) | 0.034 | 0.081 | 33.8 | 27.7 | 52.1 | 1.73 | 0.035 | - | - | - | - | - |
| 3D Infomax (Stärk et al., 2022) | 0.034 | 0.075 | 29.8 | 25.7 | 48.8 | 1.67 | 0.033 | - | - | - | - | - |
| 3D-MGP (Jiao et al., 2023) | 0.020 | 0.057 | 21.3 | 18.2 | 37.1 | 1.38 | 0.026 | - | - | - | - | - |
| Schnet (Schütt et al., 2017) | 0.033 | 0.235 | 41.0 | 34.0 | 63.0 | 1.7 | 0.033 | 14 | 19 | 14 | 14 | 73 |
| PhysNet (Unke & Meuwly, 2019) | 0.053 | 0.062 | 32.9 | 24.7 | 42.5 | 1.39 | 0.028 | 8.15 | 8.34 | 8.42 | 9.4 | 765 |
| Cormorant (Anderson et al., 2019) | 0.038 | 0.085 | 34.0 | 38.0 | 61.0 | 2.03 | 0.026 | 22 | 21 | 21 | 20 | 961 |
| DimeNet++ (Gasteiger et al., 2020) | 0.030 | 0.044 | 24.6 | 19.5 | 32.6 | 1.21 | 0.023 | 6.32 | 6.28 | 6.53 | 7.56 | 331 |
| PaiNN (Schütt et al., 2021) | 0.012 | 0.045 | 27.6 | 20.4 | 45.7 | 1.28 | 0.024 | 5.85 | 5.83 | 5.98 | 7.35 | 66 |
| EGNN (Satorras et al., 2021) | 0.029 | 0.071 | 29.0 | 25.0 | 48.0 | 1.55 | 0.031 | 11 | 12 | 12 | 12 | 106 |
| NoisyNode (Godwin et al., a) | 0.025 | 0.052 | 20.4 | 18.6 | 28.6 | 1.16 | 0.025 | 7.30 | 7.57 | 7.43 | 8.30 | 700 |
| SphereNet (Liu et al., 2022b) | 0.025 | 0.053 | 22.8 | 18.9 | 31.1 | 1.12 | 0.024 | 6.26 | 6.36 | 6.33 | 7.78 | 268 |
| ComENet (Wang et al., 2022) | 0.025 | 0.045 | 23.1 | 19.8 | 32.4 | 1.20 | 0.024 | 6.59 | 6.82 | 6.86 | 7.98 | 259 |
| SEGNN (Brandstetter et al., 2022) | 0.023 | 0.060 | 24.0 | 21.0 | 42.0 | 1.62 | 0.031 | 15 | 13 | 16 | 15 | 660 |
| EQGAT (Le et al., 2022) | 0.011 | 0.053 | 20.0 | 16.0 | 32.0 | 2.00 | 0.024 | 25 | 25 | 24 | 23 | 382 |
| LEFTNet (Du et al., 2024) | 0.011 | 0.039 | 23 | 18 | 39 | 1.19 | 0.022 | 5 | 5 | 5 | 6 | 66 |
| SaVeNet (Aykent & Xia, 2024) | 0.0085 | 0.035 | 16.6 | 15.1 | 22.7 | 1.10 | 0.021 | 4.83 | 4.74 | 4.83 | 6.10 | 49 |
| SE(3)-T (Fuchs et al., 2020) | 0.051 | 0.142 | 35.0 | 33.0 | 53.0 | - | 0.052 | - | - | - | - | - |
| TorchMD-Net (Thölke & De Fabritiis, 2022) | 0.011 | 0.059 | 20.3 | 17.5 | 36.1 | 1.84 | 0.026 | 6.15 | 6.38 | 6.16 | 7.62 | 33 |
| Equiformer (Liao & Smidt) | 0.011 | 0.046 | 15.0 | 14.0 | 30.0 | 1.26 | 0.023 | 6.59 | 6.74 | 6.63 | 7.63 | 251 |
| Transformer-M (Luo et al., 2022) | 0.037 | 0.041 | 17.5 | 16.2 | 27.4 | 1.18 | 0.022 | 9.37 | 9.41 | 9.39 | 9.63 | 75 |
| TGT (Hussain et al., 2024) | 0.025 | 0.040 | 9.9 | 9.7 | 17.4 | 1.18 | 0.020 | - | - | - | - | - |
| EquiformerV2 (Liao et al., 2024) | 0.010 | 0.050 | 14 | 13 | 29 | 1.47 | 0.023 | 6.17 | 6.49 | 6.22 | 7.57 | 186 |
| EquiformerV2+NN (Liao et al., 2024) | 0.009 | 0.039 | 12.2 | 11.4 | 24.2 | 1.21 | 0.020 | 4.34 | 4.28 | 4.24 | 5.34 | 182 |
| Geoformer (Wang et al., 2024a) | 0.010 | 0.040 | 18.4 | 15.4 | 33.8 | 1.28 | 0.022 | 4.43 | 4.41 | 4.39 | 6.13 | 27.5 |
| TetraGT | 0.017 | 0.032 | 8.5 | 8.7 | 15.6 | 1.11 | 0.019 | 4.92 | 5.11 | 4.36 | 6.07 | 35 |

## 4.1 LARGE-SCALE QUANTUM CHEMICAL PREDICTION

**PCQM4Mv2.** PCQM4Mv2, part of the OGB-LSC graph property prediction challenge, contains over 3.7 million molecules. The dataset task is to predict the HOMO-LUMO gap. The performance of the distance predictor is tuned on a random 5% subset of the training data, which we refer to as validation-3d. Experimental results, expressed as Mean Absolute Error (MAE) in meV, are presented in Table 2. We observe that the 24-layer TetraGT model achieves the best performance on the PCQM4Mv2 dataset, surpassing all baseline models and outperforming the previous state-of-the-art TGT model by 1.1 meV, demonstrating the effectiveness of our proposed approach. Notably, even without RDkit conformers as input, the 24-layer TetraGT model relying solely on 2D molecular graphs achieves comparable performance to the previous state-of-the-art TGT that utilized RDkit conformers. The gap between the 12-layer and 24-layer TetraGT suggests that effectively encoding higher-order substructures on graphs requires deeper model architectures and larger model capacities.

**Open Catalyst 2020 IS2RE.** The Open Catalyst 2020 Challenge aims to predict the adsorption energy of molecules on catalyst surfaces. We conduct experiments on the IS2RE (Initial Structure to Relaxed Energy) task. The IS2RE dataset provides initial Density Functional Theory(DFT) structures of crystals and adsorbates. Following TGT's experimental configuration, we crop/sample based on the distance to adsorbate atoms, limiting the number of atoms to a maximum of 64. IS2RE task results are presented in Table 3, expressed as MAE (in meV) and Energy within Threshold (EwT) at 20 meV. The table demonstrates that TetraGT achieves state-of-the-art (SOTA) performance across all subsets of the IS2RE evaluation dataset, both in terms of absolute values and effective proportion, without significantly increasing computational resources. This firmly establishes TetraGT as the best-performing direct method on the OC20 IS2RE task.

## 4.2 TRANSFER LEARNING

Our model learns two complementary types of knowledge on the PCQM4Mv2 dataset: the conformer predictor acquires geometric information by predicting conformations, and the task predictor learns quantum chemical properties by predicting the HOMO–LUMO gap. We evaluate how effectively these two forms of knowledge learned by TetraGT transfer to diverse downstream tasks.

**3D Downstream Tasks.** For tasks with available 3D structural information, we directly fine-tune the task predictor using precise 3D conformational data during inference. On QM9, TetraGT achieves state-of-the-art results in 5 out of 12 tasks as demonstrated in Table 4, surpassing TGT on all targets and significantly outperforming other models in HOMO ($\varepsilon_H$), LUMO ($\varepsilon_L$), and

Table 5: Results on PDBBind core set (version 2016). The evaluation metrics include Pearson's correlation coefficient (R), Mean Absolute Error (MAE), Root-Mean Squared Error (RMSE), and Standard Deviation (SD).

| Method | R↑ | MAE↓ | RMSE↓ | SD↓ |
|---|---|---|---|---|
| RF-Score Ballester & Mitchell (2010) | $0.789_{(0.003)}$ | $1.161_{(0.007)}$ | $1.446_{(0.008)}$ | $1.335_{(0.010)}$ |
| OnionNet Zheng et al. (2019) | $0.768_{(0.014)}$ | $1.078_{(0.028)}$ | $1.407_{(0.034)}$ | $1.391_{(0.038)}$ |
| GNN-DTI Lim et al. (2019) | $0.736_{(0.021)}$ | $1.192_{(0.032)}$ | $1.492_{(0.025)}$ | $1.471_{(0.051)}$ |
| DMPNN Yang et al. (2019) | $0.729_{(0.006)}$ | $1.188_{(0.009)}$ | $1.493_{(0.016)}$ | $1.489_{(0.014)}$ |
| SGCN Shi et al. (2021) | $0.686_{(0.015)}$ | $1.250_{(0.036)}$ | $1.583_{(0.033)}$ | $1.582_{(0.320)}$ |
| DimeNet Gasteiger et al. (2020) | $0.752_{(0.010)}$ | $1.138_{(0.026)}$ | $1.453_{(0.027)}$ | $1.434_{(0.023)}$ |
| CMPNN Song et al. (2020) | $0.765_{(0.009)}$ | $1.117_{(0.031)}$ | $1.408_{(0.028)}$ | $1.399_{(0.025)}$ |
| SIGN Li et al. (2021) | $0.797_{(0.012)}$ | $1.027_{(0.025)}$ | $1.316_{(0.031)}$ | $1.312_{(0.035)}$ |
| Transformer-M Luo et al. (2022) | $0.830_{(0.011)}$ | $0.940_{(0.006)}$ | $1.232_{(0.013)}$ | $1.207_{(0.007)}$ |
| TetraGT | $\mathbf{0.852}_{(0.017)}$ | $\mathbf{0.909}_{(0.012)}$ | $\mathbf{1.184}_{(0.015)}$ | $\mathbf{1.181}_{(0.010)}$ |

Table 6: Results on Peptides-func and Peptides-struct datasets.

| Model | Peptides-func | Peptides-struct |
|---|---|---|
| | Avg. Precision(%)↑ | MAE↓ |
| SAN Kreuzer et al. (2021) | $64.39_{(0.75)}$ | $0.2545_{(0.0012)}$ |
| GraphGPS Rampášek et al. (2022b) | $65.34_{(0.91)}$ | $0.2509_{(0.0014)}$ |
| MGT Geng et al. (2024) | $68.17_{(0.64)}$ | $0.2453_{(0.0025)}$ |
| DRew Gutteridge et al. (2023) | $71.50_{(0.44)}$ | $0.2536_{(0.0015)}$ |
| Graph ViT He et al. (2023) | $69.70_{(0.80)}$ | $0.2449_{(0.0016)}$ |
| GRIT Ma et al. (2023) | $69.88_{(0.82)}$ | $0.2460_{(0.0012)}$ |
| GRED Ding et al. (2023) | $71.33_{(0.11)}$ | $0.2455_{(0.0013)}$ |
| TIGT Choi et al. (2024) | $66.79_{(0.74)}$ | $0.2485_{(0.0015)}$ |
| GPNN Yang et al. (2022) | $69.55_{(0.57)}$ | $0.2454_{(0.0003)}$ |
| GSSC Huang et al. (2024) | $70.81_{(0.62)}$ | $0.2459_{(0.0020)}$ |
| TetraGT | $\mathbf{72.86}_{(0.39)}$ | $\mathbf{0.2421}_{(0.0017)}$ |

Table 7: Distance and angle prediction performance of different angle interaction mechanisms and training times on PCQM4Mv2 validation-3D set.

| | No 4th- and 5th-Order | Axial Att. | Full Att. | Tetrahe. Att. |
|---|---|---|---|---|
| Dist. Cross-Ent.(↓) | 1.204 | 1.164 | 1.179 | **1.125** |
| Angle Cross-Ent.(↓) | - | 1.310 | 1.307 | **1.231** |
| Time/Epoch(↓) | **1.00** | 1.36 | 1.43 | 1.12 |

Table 8: Ablation Study on PCQM4Mv2.

| Tetrahe. Interac. Module | Directed Cycle Loss | Hierarch. Virtual Node | Mode Distribution ($p_{Distance}, p_{Angle}$) | Val. MAE↓ (meV) |
|---|---|---|---|---|
| - | - | - | - | 73.6 |
| ✓ | - | - | - | 71.0 |
| ✓ | ✓ | - | - | 70.6 |
| ✓ | ✓ | ✓ | 1:1 | 70.2 |
| ✓ | ✓ | ✓ | 1:2 | 70.7 |
| ✓ | ✓ | ✓ | 2:1 | 69.5 |
| ✓ | ✓ | ✓ | 4:1 | **68.8** |
| ✓ | ✓ | ✓ | 8:1 | 70.1 |

HOMO–LUMO gap ($\Delta\varepsilon$) predictions—tasks that are most closely aligned with our pre-training objectives on PCQM4Mv2. On the remaining QM9 properties, TetraGT typically attains second-best or highly competitive performance. This diverse pattern is consistent with the alignment between our pre-training supervision (HOMO–LUMO gap prediction and distance/angle denoising) and the underlying physics of different QM9 properties: energy- and orbital-related quantities benefit most directly, whereas properties that depend more on long-range polarization or global shape only indirectly optimized due to their similarity angle to the pre-training task. A more detailed analysis of these task-dependent behaviors is provided in Appendix G.1. The PDBBind (version 2016) (Wang et al., 2004) results reveal TetraGT's superior binding affinity prediction capabilities, achieving state-of-the-art performance with $R = 0.852$ and MAE$= 0.909$, substantially outperforming other baselines (see Table 5). Similarly impressive results emerge from Peptides-struct (Dwivedi et al., 2022), where TetraGT achieves the lowest MAE of 0.2421 across 11 structural regression tasks, as detailed in Table 6. These consistent improvements across diverse 3D tasks demonstrate that TetraGT effectively facilitates knowledge transfer beyond the specific pre-training dataset.

**2D Downstream Tasks.** For datasets lacking 3D coordinates, we employ TetraGT's pre-trained conformer predictor as a frozen feature extractor to provide structural information. Table 6 shows TetraGT achieving 72.86% Average Precision on Peptides-func (Dwivedi et al., 2022), a 10-way functional classification task, surpassing all baseline methods by significant margins. The LIT-PCBA (Tran-Nguyen et al., 2020) drug discovery benchmark further validates this approach, with TetraGT achieving superior average ROC-AUC across 7 protein interaction prediction tasks compared to other pre-trained models, as reported in Table 1. The consistent improvements across both 2D datasets indicate that molecules with only 2D representations can also obtain effective and valuable information from the 3D data-pretrained conformer predictor of TetraGT. Detailed dataset descriptions and additional experimental analyses are provided in Appendix G.

## 4.3 ABLATION STUDY

Table 7 compares the impact of different angle interaction methods on the prediction of the interatomic distance, the prediction of the angle in conformations, and the training time. Table 8 presents an ablation study of our three main optimization designs and the ratio of distance to angle loss for the 12-layer TetraGT model. The results show that all three components positively contribute to performance, with the tetrahedral interaction module providing the most significant gains, while the Directed Cycle Angle Loss and Hierarchical Virtual Nodes further improve optimization stability and multi-level feature aggregation. Lastly, we experimented with different ratios of distance loss to angle loss and found that the model performs best when the ratio is 1:4.

## 4.4 EFFICIENCY AND SCALABILITY ANALYSIS

TetraGT's strong performance on PDBBind and Peptides datasets, which contain protein-ligand complexes and peptides with up to hundreds of atoms, already demonstrates the model's capability to handle large molecular systems. To further assess scalability with respect to molecular size, we additionally evaluate TetraGT on the OC20 IS2RE dataset; detailed experimental settings and analysis are provided in Appendix H. In terms of empirical efficiency, across PCQM4Mv2 and OC20 IS2RE, 6/12/24-layer TetraGT models consistently achieve lower MAE than UniMol+ and TGT under comparable or shorter training and inference times, while its pretraining cost on OC20 is more than three times lower than UniMol+ (33 vs. 112 A100 GPU days). These results indicate that the tetrahedral geometry-driven modules substantially enhance accuracy without incurring prohibitive computational overhead; more fine-grained efficiency analysis are given in Appendix E.

## 5 CONCLUSION

In this work, we introduce the TetraGT architecture, which directly models molecular geometric parameters (such as bond angles and torsion angles) and enables effective interactions between higher-order structures through tetrahedral attention, significantly enhancing the accuracy of molecular geometry modeling and local chirality expression. In future research, we plan to investigate dynamic representations of molecular geometric parameters in spatial stereochemistry, enabling more effective and rational geometric constraints for structural predictions.

## ACKNOWLEDGMENTS

Part of this work was done during an internship at BioMap. The work was partially done at Gaoling School of Artificial Intelligence, Beijing Key Laboratory of Research on Large Models and Intelligent Governance, Engineering Research Center of Next-Generation Intelligent Search and Recommendation, MOE, and Pazhou Laboratory (Huangpu), Guangzhou, Guangdong 510555, China. This research was supported in part by National Science and Technology Major Project (2022ZD0114802), by National Natural Science Foundation of China (No. U2241212, No. 92470128, No. 62376275). We also wish to acknowledge the support provided by the fund for building world-class universities (disciplines) of Renmin University of China, by Engineering Research Center of Next-Generation Intelligent Search and Recommendation, Ministry of Education, by Intelligent Social Governance Interdisciplinary Platform, Major Innovation & Planning Interdisciplinary Platform for the "Double-First Class" Initiative, Public Policy and Decision-making Research Lab, and Public Computing Cloud, Renmin University of China.

## 6 ETHICS STATEMENT

All data used in this study are publicly available and do not contain personally identifiable information. The research was conducted in accordance with the ethical guidelines for computational research, ensuring that all methodologies and procedures followed appropriate ethical standards. The authors declare that they have no conflicts of interest related to this study.

## 7 REPRODUCIBILITY STATEMENT

All experimental code will be made publicly available upon paper acceptance. Detailed hyperparameters and experimental configurations are provided in the Appendix F. All experiments were conducted using the same hardware configuration described in Appendix F.

## 8 LLM USAGE

Large language models (LLMs) were used for refining sentence structure, improving grammatical accuracy, and enhancing the clarity of the manuscript text. A supporting role was played by the LLMs in the manuscript's language polishing, but no scientific content, data analysis, or experimental design was generated by the LLMs.

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

## A  BROADER IMPACTS

Through our paper, we demonstrate that TetraGT can better predict molecular properties and generate precise conformations in drug development, showing its powerful applicability in practical drug discovery tasks. However, given its effectiveness, TetraGT could potentially be misused to generate harmful molecular conformations and predict toxic properties. This risk can be mitigated by comprehensively considering toxicity and other side effects in the properties of downstream tasks, or by screening out harmful molecular conformations during the generation process.

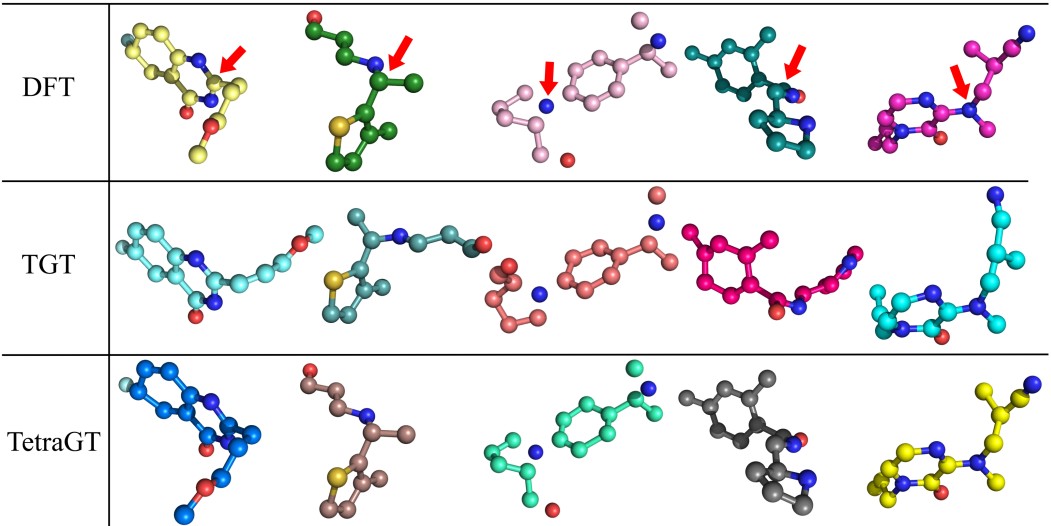

Figure 3: The ability to identify local chirality. The first row depicts DFT conformations. The second and third rows show the corresponding molecular conformations from TGT distance predictor and TetraGT conformations predictor. TetraGT can accurately generate molecules with local chirality identical to the target conformation, whereas TGT conformations, relying solely on distance matrices, exhibit deviations. Red arrows indicate atoms representing the centers of local chirality in the molecules.

## B  DENSITY FUNCTIONAL THEORY FOR MOLECULAR CONFORMATION PREDICTION

Density Functional Theory (DFT) (Kohn et al., 1996; Orio et al., 2009) is a first-principles computational method based on quantum mechanics that plays a crucial role in molecular conformation generation and property prediction. DFT describes many-electron systems through electron density rather than wave functions, significantly reducing computational complexity. Its theoretical foundation rests on the Hohenberg-Kohn theorem, which proves that all properties of a system's ground state can be uniquely determined by the electron density. In practical applications, the complex many-electron problem is transformed into more tractable single-electron problems through the Kohn-Sham equations. In molecular conformation generation, DFT can obtain precise three-dimensional conformations of molecules by solving electronic structure equations. This process includes optimizing molecular geometry, calculating bond lengths, bond angles, and dihedral angles, determining the lowest energy conformation, and predicting electron distribution within molecules. The molecular conformations generated by DFT possess high accuracy and are often used as benchmarks for the evaluation of other conformation generation methods. This high precision stems from its rigorous quantum mechanical theoretical foundation, which can accurately describe electronic effects, chemical bonding properties, and intramolecular interactions in molecules. However, DFT calculations also have limitations, such as high computational cost and difficulty in handling large molecular systems. In modern molecular design, DFT often complements machine learning methods (Schütt et al., 2017; Axelrod & Gomez-Bombarelli, 2022; Smith et al., 2020). Machine learning models can quickly predict molecular properties and initial conformations, while DFT is used to generate high-precision reference conformations and validate results. This combination leverages the advantages of both methods: the efficiency of machine learning and the high accuracy of DFT. With improvements in computational power and algorithms, DFT's applications in molecular science research will continue to expand, providing crucial support for drug design, materials development, and other fields.

## C  QUANTITATIVE ANALYSIS OF CHIRALITY PREDICTION

The conformation predictor outputs binned distances and angles under local chirality constraints, providing essential structural information for downstream task predictors. To quantitatively evaluate

TetraGT's improvement over TGT in handling local chirality, we conducted a systematic evaluation on the PCQM4M training set, which contains 3,803,453 molecules, including 1,772,922 molecules with chiral centers (46.61%). The evaluation methodology compares model-predicted 3D conformers with high-precision DFT-calculated conformers, using angular deviations around chiral centers as the assessment criterion, with a deviation threshold of $\pi/6$. Experimental results demonstrate TetraGT's superiority over the baseline TGT model across three key metrics in Table 9. In terms of bond angle MAE, TetraGT achieves 0.205 rad, a 16.6% reduction compared to TGT's 0.246 rad. For torsion angle MAE, TetraGT reaches 0.312 rad, significantly lower than TGT's 0.597 rad by 47.7%. Regarding chirality prediction accuracy, TetraGT attains 76.7%, substantially outperforming TGT's 32.5% with a 136% improvement. These quantitative results strongly validate TetraGT's excellence in modeling chiral structures, particularly in complex torsion angle prediction and overall chirality determination tasks. The substantial improvements across all metrics demonstrate the effectiveness of TetraGT's direct angle modeling approach in capturing local molecular geometry.

Table 9: Comparison of TetraGT and TGT performance on chirality prediction

| Model | Bond Angles MAE (rad) | Torsion Angles MAE (rad) | Chirality Pred (%) |
|---|---|---|---|
| TGT | 0.246 | 0.597 | 32.5 |
| TetraGT | **0.205** | **0.312** | **76.7** |

## D  THE ACCURACY OF CONFORMATION PREDICTOR IN ANGLES AND DISTANCES

To demonstrate the accuracy of TetraGT in geometric conformation prediction, we convert distances and angles to continuous unbounded values. Following the strategy employed in TGT (Hussain et al., 2024), we train two small refinement networks for distances and angles respectively. These networks accept clipped and binned values as input and output continuous, unbounded values. We train these networks using MAE loss and employ random inference to obtain the median of the output distances. We compare the accuracy of individual pairwise distances and angles on the validation-3D split of the PCQM4Mv2 dataset (i.e., data unseen during training), based on MAE, RMSE (Root Mean Square Error), and percentage errors within different thresholds as shown in Table 10 and Table 11. Our findings indicate that in terms of distances, our TetraGT predictor outperforms TGT across all metrics. Regarding angles, TetraGT significantly surpasses both RDKit and TGT in bond angle prediction and substantially leads in torsion angle prediction. This suggests that through angle constraints, TetraGT's conformation predictor can more accurately predict the underlying structure of molecules compared to the distance predictor in TGT.

Table 10: Accuracy of pairwise distances in terms of MAE↓, RMSE↓ and percent error within a threshold (EwT↑).

| Model | MAE (Å) | RMSE (Å) | EwT-0.2Å(%) | EwT-0.1Å(%) | EwT-0.05Å(%) | EwT-0.01Å(%) |
|---|---|---|---|---|---|---|
| RDKit | 0.248 | 0.541 | 73.33 | 66.65 | 56.90 | 26.79 |
| TGT + Refiner | 0.152 | 0.378 | 80.53 | 75.68 | 70.80 | 54.54 |
| TetraGT + Refiner | **0.127** | **0.322** | **87.32** | **79.15** | **75.31** | **58.29** |

Table 11: Accuracy of bond angles and torsion angles in terms of MAE↓, RMSE↓ and percent error within a threshold (EwT↑).

| **Model** | Bond Angles | | | Torsion Angles | | |
|---|---|---|---|---|---|---|
| | MAE (rad) | RMSE (rad) | EwT-$\pi/16$ rad (%) | MAE (rad) | RMSE (rad) | EwT-$\pi/16$ rad (%) |
| RDKit | 0.239 | 0.575 | 71.43 | 0.694 | 1.145 | 33.62 |
| TGT + Refiner | 0.225 | 0.431 | 76.26 | 0.563 | 0.713 | 41.89 |
| TetraGT + Refiner | **0.185** | **0.362** | **83.50** | **0.306** | **0.467** | **62.31** |

## E    EFFICIENCY ANALYSIS OF TETRAGT VERSUS BASELINE MODELS

### E.1    PCQM4Mv2

Table 12 presents a comprehensive comparison of TetraGT against state-of-the-art molecular pre-training methods, Unimol+ and TGT, across different model scales, showing parameter counts, computational complexity, experimental performance on the PCQM4Mv2 dataset, and training/inference times. Based on experimental results, we comprehensively analyze TetraGT's method from both efficiency and effectiveness perspectives. Regarding computational complexity, where N represents the number of atoms and w denotes the local sampling window size, TetraGT with local sampling achieves $O(wN^2)$ complexity for standard atom and pair embedding interactions through restricting attention to w nearest neighbors, compared to the original $O(N^3)$ complexity. In typical molecules, the number of bond angles ranges from 1.5N to 2N, and torsion angles from N to 2N. With local sampling applied to these higher-order interactions, the additional computational complexity becomes $O(wN)$, yielding an overall complexity of $O(wN^2) + O(wN) = O(wN^2)$. On the large-scale PCQM4Mv2 dataset, TetraGT demonstrates an excellent balance between performance and computational efficiency. We systematically analyzed the trade-off between model scale and performance. Results show that 6-layer TetraGT (60M parameters) achieves an MAE of 69.4 meV, comparable to 18-layer Unimol+ (77M parameters) at 69.3 meV, while significantly reducing training time (approximately 14 days versus 40 days using A100 GPU). As model layers increase, 24-layer TetraGT (215M parameters) reduces MAE to 66.2 meV, significantly outperforming 24-layer TGT (203M parameters, 67.1 meV MAE). Notably, 12-layer TetraGT (127M parameters) maintains competitive performance (68.1 meV MAE) while reducing training time from 38 to 20 GPU days, with corresponding inference time reduction. While the efficiency improvement from local sampling is less pronounced on PCQM4Mv2 due to its relatively small molecular sizes (average 14 atoms) and TetraGT's additional higher-order structural interactions, this strategy is crucial for scalability—it enables TetraGT to successfully process large biomolecular datasets like PDBbind and Peptides, which would be computationally intractable without local sampling. These results indicate that TetraGT architecture is competitive even at smaller scales and can better leverage its structural modeling advantages as parameter count increases.

### E.2    OPEN CATALYST 2020 IS2RE

Table 13 presents a comprehensive evaluation of TetraGT against both pre-trained and non-pre-trained methods on the OC20 dataset, focusing on computational efficiency and model performance. Based on experimental results, we analyze TetraGT's capabilities from multiple perspectives. Regarding computational efficiency, TetraGT demonstrates competitive inference and fine-tuning times compared to non-pre-training methods. Specifically, TetraGT's fine-tuning duration (220 minutes) aligns well with established models such as DimeNet++ (230 minutes), GemNet-T (200 minutes), and SphereNet (290 minutes). While ComENet exhibits faster training speed (20 minutes), TetraGT achieves substantially superior performance metrics, with energy MAE of 397.7 meV versus 588.8 meV and FwT of 9.14% versus 3.56%, validating the effectiveness of our pre-training strategy. In comparison with other pre-trained methods, TetraGT shows remarkable efficiency improvements while maintaining performance advantages. Compared to TGT, despite incorporating additional angular information and direct angle modeling mechanisms, TetraGT maintains similar training efficiency (approximately 34 days versus 32 days using A100 GPU) while achieving superior performance. Notably, compared to Uni-Mol+, TetraGT achieves better performance metrics while significantly reducing pre-training time (34 days versus 112 days using A100 GPU), demonstrating an optimal balance between computational efficiency and model effectiveness.

## F    EXPERIMENTAL DETAILS

The model is implemented using the PyTorch (Paszke et al., 2019) library. We perform mixed-precision training on 2 nodes, each equipped with 8 NVIDIA Tesla A100 GPUs (80GB RAM/GPU) and 16-core 2.6GHz Intel Xeon CPUs (320GB RAM per node). The hyperparameters used for each dataset are presented in Table 14. For PCQM4Mv2 and OC20 we list the hyperparameters for both the conformation and the task predictor models and both training and finetuning. For QM9, we only list the hyperparameters for finetuning. For MOLPCBA, LIT-PCBA, and MOLHIV we only show

Table 12: Comparison of performance and efficiency metrics on PCQM4Mv2

| Model | # param. | Complexity | # layers | MAE (meV) | Training Time | Inference Time |
|---|---|---|---|---|---|---|
| Unimol+ | 27.7M | $O(N^3)$ | 6 | 71.4 | ~12 A100 GPU day | ~19 A100 GPU min |
| Unimol+ | 52.4M | $O(N^3)$ | 12 | 69.6 | ~42 A100 GPU day | ~32 A100 GPU min |
| Unimol+ | 77M | $O(N^3)$ | 18 | 69.3 | ~40 A100 GPU day | ~56 V100 GPU min |
| TGT | 116M | $O(N^3)$ | 12 | 70.9 | ~18 A100 GPU day | ~27 A100 GPU min |
| TGT | 203M | $O(N^3)$ | 24 | 67.1 | ~32 A100 GPU day | ~40 A100 GPU min |
| TetraGT | 60M | $O(wN^2)$ | 6 | 69.3 | ~10 A100 GPU day | ~16 A100 GPU min |
| TetraGT | 127M | $O(wN^2)$ | 12 | 68.1 | ~18 A100 GPU day | ~29 A100 GPU min |
| TetraGT | 215M | $O(wN^2)$ | 24 | **65.9** | ~34 A100 GPU day | ~33 A100 GPU min |

Table 13: Comparison of performance and efficiency metrics on OC20

| Model | Pretraining Time | Train Time | Inference Time | Avg. Energy MAE (meV) ↓ | Avg. FwT (%) ↑ |
|---|---|---|---|---|---|
| CGCNN | - | 18 min | 1 min | 658.5 | 2.82 |
| SchNet | - | 10 min | 1 min | 666.0 | 2.65 |
| DimeNet++ | - | 230 min | 4 min | 621.7 | 3.42 |
| GemNet-T | - | 200 min | 4 min | 638.2 | 3.38 |
| SphereNet | - | 290 min | 5 min | 602.3 | 3.64 |
| ComENet | - | 20 min | 1 min | 588.8 | 3.56 |
| Unimol+ | 112 A100 GPU days | 340 min | 8 min | 408.8 | 8.61 |
| TGT | 32 A100 GPU days | 200 min | 5 min | 403.0 | 8.82 |
| TetraGT | 33 A100 GPU days | 220 min | 5 min | **397.7** | **9.14** |

the hyperparameters for training from scratch. The missing hyperparameters do not apply to the corresponding dataset or model. For QM9 no secondary distance and angle denoising objective is used. For LIT-PCBA, 0 triplet interaction heads indicate that an EGT is used without any triplet interaction module.

To provide the conformation predictor with initial 3D information, we utilize RDKit (Landrum, 2013) to extract 3D coordinates and apply MM Force Field Optimization (Halgren, 1996). Due to the absence of Ground Truth 3D coordinates in the PCQM4Mv2 validation set, we randomly divide the training set into train-3D and validation-3D splits, with the latter containing 5% of the training data. Hyperparameters of the conformation predictor are fine-tuned by monitoring the average cross-entropy loss of binned distance and angle prediction on the validation-3D split, which is found to be a good indicator of downstream performance. The input noise level is adjusted by evaluating the finetuned performance on the validation set. We get the best results by using an average of 50 sample predictions during stochastic inference. Other training configurations not mentioned are based on TGT (Hussain et al., 2024) https://github.com/shamim-hussain/tgt (MIT license).

# G ADDITIONAL RESULTS AND ANALYSES

## G.1 QM9

We present the comprehensive evaluation results on the QM9 dataset across all 12 prediction tasks in Table 4. Overall, TetraGT demonstrates strong and consistent predictive capabilities across diverse molecular properties. In particular, it achieves state-of-the-art performance on several energy-related metrics, including HOMO energy ($\varepsilon_H$: 8.5), LUMO energy ($\varepsilon_L$: 8.7), energy gap ($\Delta\varepsilon$: 15.6), and heat capacity ($C_v$: 0.019). On the remaining targets where it does not rank first, TetraGT typically attains second-best or competitive performance close to the top of the benchmark (e.g., $C_v$: 0.020, matching TGT's performance), indicating that its overall performance level is comparable to the strongest existing methods rather than being specialized to a narrow subset of tasks. For optical and quantum properties such as $\alpha$ and ZPVE, TetraGT remains highly competitive, and for thermodynamic quantities ($U_0$, $U$, $H$, $G$) and geometric features ($R^2$), it surpasses previous pre-trained approaches including Transformer-M.

Table 14: Hyperparameters for each dataset.

| Hyperparameters | PCQM4Mv2 Conf. Pred. | PCQM4Mv2 Task Pred. | OC20 Conf. Pred. | OC20 Task Pred. | QM9 Task Pred. | PDBbind Task Pred. | LIT-PCBA Task Pred. | Peptides* Task Pred. |
|---|---|---|---|---|---|---|---|---|
| Sampl. Window Size | 12 | 12 | 15 | 15 | 10 | 64 | 10 | 32 |
| # Layers | 24 | 24 | 24 | 14 | 24 | 24 | 8 | 24 |
| Node Embed. Dim | 768 | 768 | 768 | 768 | 768 | 768 | 1024 | 768 |
| Edge Embed. Dim | 256 | 256 | 256 | 512 | 256 | 256 | 256 | 256 |
| Angle Embed. Dim | 128 | 128 | 128 | 256 | 128 | 128 | 128 | 128 |
| # Attn. Heads | 64 | 64 | 64 | 64 | 64 | 64 | 64 | 64 |
| # Triplet Heads | 16 | 16 | 16 | 16 | 16 | 16 | 0 | 16 |
| Node FFN Dim. | 768 | 768 | 1536 | 768 | 768 | 768 | 2048 | 768 |
| Edge FFN Dim. | 256 | 256 | 512 | 512 | 256 | 256 | 512 | 256 |
| Angle FFN Dim. | 128 | 128 | 256 | 256 | 128 | 128 | 256 | 128 |
| Max. Hops Enc. | 32 | 32 | - | - | 32 | 32 | 32 | 32 |
| Activation | GELU | GELU | GELU | GELU | GELU | GELU | GELU | GELU |
| Input Dist. Enc. | RBF | RBF | Fourier | Fourier | RBF | RBF | RBF | RBF |
| Source Dropout | 0.3 | 0.3 | 0.3 | 0.3 | 0.3 | 0.3 | 0.3 | 0.3 |
| Triplet Dropout | 0.0 | 0.0 | 0.1 | 0.0 | 0.0 | 0.1 | 0.0 | 0.0 |
| Path Dropout | 0.2 | 0.2 | 0.2 | 0.1 | 0.2 | 0.1 | 0.1 | 0.1 |
| Node Activ. Dropout | 0.1 | 0.1 | 0.1 | 0.1 | 0.1 | 0.1 | 0.1 | 0.1 |
| Edge Activ. Dropout | 0.1 | 0.1 | 0.1 | 0.1 | 0.1 | 0.1 | 0.1 | 0.1 |
| Angle Activ. Dropout | 0.1 | 0.1 | 0.1 | 0.1 | 0.1 | 0.1 | 0.1 | 0.1 |
| Input 3D Noise | - | 0.2 | - | 0.6 | 0.0 | 0.0 | - | - |
| Input Noise Smooth. | - | 1.0 | - | 1.0 | 0.0 | 0.0 | - | - |
| Optimizer | Adam | Adam | Adam | Adam | Adam | Adam | Adam | Adam |
| Batch Size | 1024 | 2048 | 256 | 256 | - | - | 1024 | 128 |
| Max. LR | 0.001 | 0.0015 | 0.001 | 0.001 | - | - | $5 \times 10^{-4}$ | $2 \times 10^{-4}$ |
| Min. LR | $10^{-6}$ | $10^{-6}$ | 0.001 | $10^{-6}$ | - | - | $5 \times 10^{-7}$ | $10^{-6}$ |
| Warmup Steps | 30000 | 20000 | 8000 | 16000 | - | - | 600 | 10000 |
| Total Training Steps | 60000 | 350000 | 30000 | 100000 | - | - | 1200 | 30000 |
| Grad. Clip. Norm | 5.0 | 5.0 | 5.0 | 5.0 | 5.0 | 5.0 | 2.0 | 5.0 |
| Conf. Loss Weight | - | 0.1 | - | 3.0 | 0.0 | 0.0 | 0.1 | 0.1 |
| # Angle Bins | 256 | 512 | 256 | 512 | - | - | 512 | 512 |
| # Dist. Bins | 256 | 512 | 256 | 512 | - | - | 512 | 512 |
| Dist. Bins Range | 8 | 8 | 16 | 16 | - | - | 8 | 8 |
| FT Batch Size | - | 2048 | - | 1024 | 2048 | 64 | - | 128 |
| FT Warmup Steps | - | 3000 | - | 0 | 3000 | 1000 | - | 2000 |
| FT Total Steps | - | 50000 | - | 12000 | 150000 | 20000 | - | 30000 |
| FT Max. LR | - | $2 \times 10^{-4}$ | - | $10^{-5}$ | $2 \times 10^{-4}$ | $2 \times 10^{-4}$ | - | $2 \times 10^{-4}$ |
| FT Min. LR | - | $10^{-6}$ | - | $10^{-5}$ | $10^{-6}$ | $10^{-6}$ | - | $10^{-6}$ |
| FT Conf. Loss Weight | - | 0.1 | - | 2.0 | 0.1 | 0.1 | - | 0.1 |

*The hyperparameters in the Peptides column represent both Peptides-func (2D) and Peptides-struct (3D). They share model hyperparameters, with the training hyperparameters in the second and third sections applying to Peptides-func (2D), and the fine-tuning hyperparameters in the last section applying to Peptides-struct (3D).

The heterogeneous performance across different QM9 properties can be better understood by examining the alignment between our pre-training objectives and the underlying physics of each target. TetraGT is pre-trained on PCQM4Mv2 with a dual objective that (i) predicts the HOMO–LUMO gap and (ii) denoises inter-atomic distances and angular variables. This setup primarily encourages the model to learn a holistic description of the electronic structure—especially the frontier orbitals—together with a geometrically consistent 3D representation. As a result, properties in QM9 whose behavior is tightly coupled to frontier orbital energies and global electronic configurations (namely $\varepsilon_H$, $\varepsilon_L$, and $\Delta\varepsilon$, as well as some energy-related thermodynamic quantities) benefit most directly from the pre-trained representations, and indeed TetraGT exhibits the largest gains on these targets. By contrast, properties such as the dipole moment $\mu$, polarizability $\alpha$, ZPVE, or $R^2$ depend more heavily on long-range charge distribution, polarization effects, and global shape fluctuations, which are only indirectly constrained by our current pre-training objective. In this sense, the supervision signal derived from the HOMO–LUMO gap induces a natural bias towards metrics with similar distributions and physical dependencies, leading to excellent but not universally dominant performance on all 12 tasks.

We also note that architectural design choices further shape this behavior when compared to highly specialized equivariant energy models. Methods such as EquiformerV2+NN adopt strict SE(3)-equivariant architectures tailored to continuous 3D coordinates and their derivatives (forces), which provides a very strong inductive bias for small-molecule energy and force prediction. In contrast, TetraGT is designed as a general-purpose geometric representation learner: it introduces explicit

Table 15: LIT-PCBA results in terms of ROC-AUC↑ (%).

| | ALDH1 | FEN1 | GBA | KAT2A | MAPK1 | PKM2 | VDR | Average |
|---|---|---|---|---|---|---|---|---|
| No. active | 7,168 | 369 | 166 | 194 | 308 | 546 | 884 | |
| No. inactive | 137,965 | 355,402 | 296,052 | 348,548 | 62,629 | 245,523 | 355,388 | |
| NaiveBayes (Webb et al., 2010) | 69.3 | 87.6 | 70.9 | 65.9 | 68.6 | 68.4 | 80.4 | 73.0 |
| SVM (Hearst et al., 1998) | 76.0 | 87.7 | 77.8 | 61.2 | 66.5 | 75.3 | 69.7 | 73.4 |
| RandomForest (Breiman, 2001) | 74.1 | 65.7 | 59.9 | 53.7 | 57.9 | 58.1 | 64.4 | 62.0 |
| XGBoost (Chen & Guestrin, 2016) | 75.0 | 88.8 | 83.0 | 50.0 | 59.3 | 73.7 | 78.2 | 72.6 |
| GCN (Kipf & Welling, 2016) | 73.0 | 89.7 | 73.5 | 62.1 | 66.8 | 63.6 | 77.3 | 72.3 |
| GAT (Velickovic et al., 2017) | 73.9 | 88.8 | 77.6 | 66.2 | 69.7 | 72.4 | 78.0 | 75.2 |
| FP-GNN (Cai et al., 2022) | 76.6 | 88.9 | 75.1 | 63.2 | **77.1** | 73.2 | 77.4 | 75.9 |
| EGT (Hussain et al., 2022) | 78.7$_{(2)}$ | 92.9$_{(1)}$ | 75.4$_{(4)}$ | 72.8$_{(1)}$ | 75.3$_{(3)}$ | 76.5$_{(2)}$ | 80.7$_{(2)}$ | 78.9 |
| GEM (Fang et al., 2022) | 77.2$_{(1)}$ | 91.4$_{(2)}$ | 82.1$_{(2)}$ | 74.0$_{(1)}$ | 71.0$_{(2)}$ | 74.6$_{(2)}$ | 78.5$_{(1)}$ | 78.4 |
| GEM-2 (Liu et al., 2022a) | 80.2$_{(0.2)}$ | 94.5$_{(0.3)}$ | 85.6$_{(2)}$ | 76.3$_{(1)}$ | 73.3$_{(1)}$ | 78.2$_{(0.4)}$ | 82.3$_{(0.5)}$ | 81.5 |
| EGT+RDKit (Hussain et al., 2024) | 80.2$_{(0.2)}$ | 95.2$_{(0.3)}$ | 84.5$_{(4)}$ | 74.3$_{(1)}$ | 73.5$_{(1)}$ | 78.0$_{(0.2)}$ | 82.8$_{(0.3)}$ | 81.2 |
| TGT (Hussain et al., 2024) | 80.6$_{(0.3)}$ | 95.5$_{(0.3)}$ | 84.4$_{(3)}$ | 74.6$_{(2)}$ | 74.3$_{(0.7)}$ | 78.4$_{(0.2)}$ | 82.9$_{(0.3)}$ | 81.5 |
| TetraGT | **81.2**$_{(0.3)}$ | **95.7**$_{(0.4)}$ | **85.6**$_{(3)}$ | 75.8$_{(2)}$ | 76.1$_{(0.8)}$ | **79.1**$_{(0.4)}$ | **83.4**$_{(0.3)}$ | **82.4** |

bond-angle and torsion-angle tokens, employs tetrahedral interaction mechanisms to encode local geometry and chirality, and uses hierarchical virtual nodes to aggregate information from atoms, bonds, angles, and torsions. This design is intended to support transfer across a wide range of tasks and scales (PCQM4Mv2, OC20 IS2RE, PDBBind, Peptides-struct, Peptides-func, LIT-PCBA), rather than being exhaustively tuned for each individual scalar property in QM9. For fairness and generality, we use a unified pre-training objective and architecture across benchmarks, without introducing task-specific loss functions or architectural variants for different QM9 targets. Inevitably, this entails a trade-off between broad generalization and task-specific optimization: TetraGT attains state-of-the-art or near–state-of-the-art performance on most properties, especially those aligned with its pre-training supervision, while some highly specialized equivariant models can still be marginally better on a subset of QM9 metrics.

In Table 4, we categorize compared methods into three distinct groups for clarity. The first group comprises pre-trained GNN methods, including GraphMVP, GEM, 3D Infomax, and 3D-MGP. The second group consists of directly trained GNN methods, spanning from SchNet through SaVeNet. The third group encompasses Transformer-based methods from SE(3)-Transformer through TetraGT; for this family, we do not distinguish between pre-trained and non-pre-trained variants because they are all trained on large-scale datasets, and our focus is on contrasting their architectural inductive biases (equivariant energy models vs. our angle-driven tetrahedral representation) rather than the source of supervision alone.

## G.2 Comprehensive Analysis on PDBBind Core Set

The PDBBind core set (version 2016) is a rigorously curated benchmark containing 290 protein-ligand complexes selected from the PDBBind refined set based on stringent quality criteria. Each complex includes high-resolution crystal structures (typically < 2.5 Å) with experimentally determined binding affinities (Kd, Ki, or IC50 values) that span approximately 10 log units from picomolar to millimolar range. The dataset provides complete 3D coordinates for both protein and ligand molecules, along with binding affinity values converted to pKd/pKi (-log10 of dissociation/inhibition constant in molar units) for consistency. The task is to predict the binding affinity value given the 3D structure of the protein-ligand complex, making it a fundamental benchmark for structure-based drug design. The core set is specifically designed to test generalization capability, with proteins clustered by 90% sequence similarity and only one representative selected from each cluster, ensuring diverse protein families and binding modes are represented.

TetraGT achieves a Pearson correlation coefficient of 0.852±0.017 on this benchmark, representing a significant advancement over previous state-of-the-art methods. The improvement over Transformer-

M (R=0.830±0.011), which previously held the best performance, demonstrates that our tetrahedral geometric constraints provide more effective inductive bias than standard transformer architectures for molecular binding prediction. More importantly, the reduction in mean absolute error from 0.940 to 0.909 meV indicates TetraGT's enhanced capability in making accurate quantitative predictions, which is crucial for practical applications in lead optimization where small differences in binding affinity can determine therapeutic efficacy.

When compared to classical scoring functions, TetraGT shows even more dramatic improvements. RF-Score, which relies on carefully engineered features based on atom-pair distance counts, achieves R=0.789, while OnionNet, using multiple layers of intermolecular contacts, reaches R=0.768. The substantial gap between these methods and TetraGT illustrates the power of learned representations that can capture complex geometric patterns beyond predefined feature sets. Among GNN-based approaches, the progression from early methods like GNN-DTI (R=0.736) to more sophisticated architectures like SIGN (R=0.797) shows steady improvement, yet TetraGT's performance leap suggests that explicitly modeling tetrahedral geometry provides crucial structural information that general graph convolutions miss.

The stability of TetraGT's predictions, evidenced by the lowest standard deviation across all metrics (SD=1.181±0.010), is particularly noteworthy. This consistency across diverse protein families and ligand chemotypes indicates that the model has learned robust representations that generalize well beyond the training distribution, a critical requirement for virtual screening applications where novel chemical scaffolds must be evaluated reliably.

### G.3 DETAILED ANALYSIS ON PEPTIDES BENCHMARKS

The Peptides-func and Peptides-struct datasets from the Long Range Graph Benchmark (LRGB) comprise 15,535 peptides ranging from 2 to 50 amino acids in length, derived from the SATPdb database of therapeutic peptides. Each peptide is represented as a molecular graph where nodes correspond to atoms and edges represent chemical bonds, with node features encoding atom types (9 categories) and edge features encoding bond types (3 categories). The datasets share the same peptide molecules but differ fundamentally in their tasks and available information. Peptides-func is a multi-label classification task predicting 10 binary functional properties: antimicrobial, antibacterial, antiviral, antifungal, anticancer, anti-HIV, antimalarial, antiparasitic, antimycobacterial, and cell-penetrating capabilities. Importantly, this dataset provides only 2D molecular graphs without experimental 3D coordinates, requiring methods to infer structural information from connectivity alone. In contrast, Peptides-struct is a regression task predicting 11 continuous structural properties computed from peptide 3D conformations: three principal components of mass inertia, three principal components of valence inertia (hydrogen distribution), three geometric axis lengths, sphericity, and plane distance. This dataset includes the actual 3D coordinates, allowing direct utilization of spatial information. Both datasets are split into train (70%), validation (15%), and test (15%) sets with stratified sampling to ensure balanced property distributions.

For Peptides-func, which lacks experimental 3D coordinates, TetraGT's approach of using a pretrained conformer predictor to generate approximate 3D features proves highly effective, achieving 72.86±0.39% Average Precision. This surpasses the previous best DRew model (71.50±0.44) by a meaningful margin, particularly impressive given the multi-label nature of the task where peptides can simultaneously exhibit multiple therapeutic properties. The improvement over GraphGPS (65.34±0.91%) and MGT (68.17±0.64%) further demonstrates that TetraGT's geometric message passing, even with predicted conformations, captures functionally relevant structural motifs more effectively than standard graph transformers. The ability to accurately predict diverse functional properties from molecular structure alone has important implications for therapeutic peptide discovery, where rapid screening of large peptide libraries is essential.

For Peptides-struct, where actual 3D structural information is available, TetraGT achieves an MAE of 0.2421±0.0017, outperforming all baseline methods including Graph ViT/MLP-Mixer (0.2449±0.0016), which previously held the best performance. The consistent improvement across all 11 structural properties suggests that TetraGT's tetrahedral representations naturally align with the geometric nature of these prediction targets. The model's ability to accurately predict properties like inertia components and geometric dimensions indicates a deep understanding of 3D molecular

geometry, essential for applications requiring precise structural modeling such as peptide docking or conformational analysis.

The performance gap between methods is particularly revealing when comparing approaches with different architectural philosophies. Traditional graph neural networks like SAN (Peptides-func: 64.39%, Peptides-struct: 0.2545) struggle with both tasks, likely due to limited expressivity in capturing long-range dependencies common in peptide structures. More recent architectures incorporating attention mechanisms show improvement, with GRIT achieving 69.88% on functional prediction and 0.2460 on structural regression. However, these gains remain incremental compared to TetraGT's substantial improvements, suggesting that generic architectural enhancements alone are insufficient without appropriate geometric inductive biases.

An interesting observation emerges when comparing performance patterns across the two datasets. Methods that excel on Peptides-func don't necessarily maintain their relative performance on Peptides-struct. For instance, DRew shows strong functional classification (71.50%) but moderate structural prediction (0.2536), while Graph ViT/MLP-Mixer shows the opposite pattern. TetraGT is unique in achieving top performance on both tasks, indicating its representations capture both functional and structural aspects effectively. This dual capability is particularly valuable for drug discovery applications where understanding both molecular function and conformation is essential.

### G.4  LIT-PCBA

We also show a breakdown of the LIT-PCBA results for the individual protein targets in Table 15. Notice that, TetraGT outperforms other models in ALDH1, FEN1, PKM2, VDR and GBA. Despite the low number of positive samples, TetraGT ranked second among all models on KAT2A and MAPK1, surpassing TGT (Hussain et al., 2024) on all proteins target. We can analyze why TetraGT shows slightly lower performance on KAT2A, and MAPK1 compared to some other methods. KAT2A and MAPK1 both have limited active samples (194 and 308 respectively) with significant class imbalance. The performance differences are relatively small - for KAT2A, TetraGT achieves 75.8% compared to GEM-2's 76.3%, and for MAPK1, TetraGT's 76.1% is the best performers. These marginal differences might be attributed to the specific structural characteristics of these proteins and the extreme class imbalance in their datasets, which could potentially benefit from more specialized handling of imbalanced data during model training.

In Tables 1 and 15, we present three groups of methods. The first group consists of traditional machine learning methods (NaiveBayes, SVM, RandomForest, XGBoost). The second group consists of directly trained GNNs (GCN, GAT, FP-GNN). The third group consists of pre-trained deep learning methods from EGT through TGT.

### G.5  PCQM4Mv2

In Table 2, we organize methods into three groups. The first group represents earlier methods, ranging from MLP-Fingerprint through GPS++. The second group includes current state-of-the-art methods (Transformer-M, Uni-Mol+, TGT) that incorporate 3D conformation perturbation and denoising prediction. The final group consists solely of our proposed TetraGT method.

### G.6  OC20

For Table 3, methods are divided into two main categories. The first group encompasses GNN methods from SchNet through GNS+NN, while the second group includes Transformer-based methods from Graphormer-3D through TGT.

## H  SCALABILITY ANALYSIS

Our comprehensive evaluation demonstrates TetraGT's robust performance across molecular systems of vastly different scales, from small organic molecules in PCQM4Mv2 (mean: 15 atoms) and MolHIV/MolPCBA (mean: 26 atoms) to large protein-ligand complexes in PDBBind (500-1000 atoms in binding sites) and complex catalytic systems in OC20 (approximately 80 atoms). TetraGT

achieves state-of-the-art performance consistently across all these benchmarks, indicating excellent scalability without architecture-specific limitations.

To systematically evaluate scalability, we conducted detailed experiments on the OC20 IS2RE dataset by grouping molecules into size segments and analyzing performance variations. Table 16 presents the Energy MAE across different molecular size ranges, demonstrating remarkably stable performance with MAE values fluctuating within a narrow band of 22 meV (354.7-377.0 meV) despite the molecular size nearly tripling from smallest to largest categories. This consistency confirms that tetrahedral message passing does not suffer from size-dependent degradation commonly observed in graph neural networks.

Table 16: Performance analysis of TetraGT model on OC20 IS2RE dataset across various molecular sizes.

| Atom_num | Energy MAE (meV) |
|---|---|
| 36-45 | 363.7 |
| 46-55 | 376.1 |
| 56-65 | 354.7 |
| 66-75 | 377.0 |
| 76-85 | 371.2 |
| 86-95 | 368.9 |
| 96-105 | 364.5 |

The absence of monotonic error trends with increasing molecular size indicates that TetraGT effectively captures both local and global geometric patterns regardless of system scale. The slight variations in MAE across different ranges likely reflect inherent complexity differences in molecular compositions rather than size-dependent model limitations. Notably, the optimal performance at 56-65 atoms aligns well with typical organic adsorbate sizes in catalysis applications, while maintaining competitive accuracy for both smaller and larger systems.

While TetraGT's architecture poses no inherent theoretical limitations on molecular size, practical applications face two primary constraints. The first stems from GPU memory capacity when processing 3D conformer data, which defines maximum processable system size. This constraint becomes particularly relevant for large biomolecular systems requiring extensive memory allocation. The second challenge arises from computational complexity scaling, as the number of higher-order structures (bond angles and torsion angles) grows quadratically with system size, potentially causing attention mechanisms to suffer from averaging effects across expanding interaction spaces.

The demonstrated scalability across diverse molecular scales, combined with consistent performance stability, positions TetraGT as a versatile tool for broad molecular modeling applications. The model's ability to maintain high accuracy from small drug-like molecules to large biomolecular complexes without size-specific modifications validates the tetrahedral message passing design as an effective balance between computational efficiency and representational power. Future exploration of the proposed optimization strategies may further extend TetraGT's applicability to even larger biochemical systems while maintaining practical computational requirements.

# I  ADDITIONAL DETAILS ABOUT RELATED WORKS

**Molecular Property Prediction** The remarkable performance of message-passing GNNs in predicting molecular properties has inspired a new generation of geometric and physics-aware neural networks, which maintain invariance or equivariance under 3D rotational and translational transformations. Early developments in this direction include SchNet (Schütt et al., 2017) and DimeNet (Gasteiger et al., 2020), which pioneered the use of distance-based convolution approaches. The field further evolved with the introduction of spherical methodologies, as exemplified by Gem-Net (Gasteiger et al., 2021), SphereNet (Liu et al., 2022b), ComENet (Wang et al., 2022), LEFT-Net (Du et al., 2024), and SAVENet (Aykent & Xia, 2024), each incorporating various forms of angular information. This architectural evolution ultimately led to more sophisticated equivariant transformer designs, including Equiformer (Liao & Smidt), EquiformerV2 (Liao et al., 2024), TorchMD-Net (Thölke & De Fabritiis, 2022), and Geoformer (Wang et al., 2024a), which generalized

the concept of equivariant aggregation. While these advances have significantly improved molecular representation learning, our work proposes a fundamentally different paradigm for modeling higher-order structures. Recent models like QuinNet (Wang et al., 2024c) and ViSNet (Wang et al., 2024b) have introduced four or five-atom interactions to enhance model expressiveness and accuracy. However, these methods primarily focus on local representations of atomic nodes and chemical bonds, capturing higher-order features implicitly through combinatorial operations between atom-level tokens. In contrast, our approach transforms higher-order graph structures into independent token representations, enabling direct learning and representation of structural patterns in molecules. This innovation is particularly crucial for model interpretability and effective utilization of expert prior knowledge. From an information propagation perspective, traditional methods require higher-order structural information (such as four-body and five-body interactions) to propagate gradually along the graph topology, creating significant information bottlenecks. As demonstrated in TGT research, even information exchange between adjacent embeddings faces restrictions. Our method addresses these limitations through direct structural token representation, not only avoiding these bottlenecks but also enabling efficient access and utilization of key higher-order information by all graph nodes, thereby providing a more effective framework for learning molecular structural information.

## J  SENSITIVITY ANALYSIS OF THE WINDOW SIZE $w$

In this section, we provide a systematic analysis of the local sampling window size $w$. In TetraGT, the tetrahedral interaction uses a local window $w$ to select neighboring angle/torsion tokens that are allowed to participate in attention. If $w$ is too small, in principle this may lead to "under-coverage" of important geometric couplings, potentially degrading both conformer prediction and property prediction. To examine this, we conduct experiments on the OC20 IS2RE dataset (in-distribution validation split), varying $w \in \{5, 10, 15, 20\}$, and study both overall performance and its dependence on molecular size.

We first evaluate, for different $w$, the conformer prediction errors (MAE of pairwise distances, bond angles, and torsion angles), the property prediction error (Energy MAE), and the relative training time per epoch (normalized such that the setting $w = 10$ corresponds to 1.0). The results are summarized in Table 17. We observe that increasing $w$ from 5 to 10 leads to clear improvements in both geometric and energy prediction: for example, Energy MAE decreases from 382.4 meV to 373.2 meV, while bond-angle MAE decreases from 0.297 rad to 0.228 rad and torsion-angle MAE from 0.387 rad to 0.316 rad. However, once $w \geq 10$, the additional gains become very limited: increasing $w$ from 10 to 15 or 20 only reduces Energy MAE by about 0.6–1.1 meV, whereas the training time per epoch increases from 1.0 to 1.52 and 2.83, respectively. This indicates that, beyond a moderate threshold (e.g., $w = 10$), TetraGT is quite robust to the choice of $w$; we do not observe noticeable accuracy degradation that could be attributed to local under-coverage, while overly large $w$ mainly increases computational cost with rapidly diminishing returns.

Table 17: Effect of different window sizes $w$ on conformer prediction and energy prediction (OC20 IS2RE, in-distribution validation).

| $w$ | Distance MAE (Å) | Bond-angle MAE (rad) | Torsion-angle MAE (rad) | Energy MAE (meV) | Rel. time / epoch |
|---|---|---|---|---|---|
| 5 | 0.191 | 0.297 | 0.387 | 382.4 | 0.76 |
| 10 | 0.168 | 0.228 | 0.316 | 373.2 | 1.00 |
| 15 | 0.163 | 0.221 | 0.313 | 372.6 | 1.52 |
| 20 | 0.157 | 0.217 | 0.309 | 372.1 | 2.83 |

To further investigate whether $w$ needs to scale with molecular size, we stratify the OC20 IS2RE in-distribution validation set by the number of atoms $N$ into four ranges: $N \leq 45$, $45 < N \leq 65$, $65 < N \leq 85$, and $85 < N \leq 105$. For each range, we report the Energy MAE under different $w$, as shown in Table 18. Across all size ranges, increasing $w$ from 5 to 10 consistently yields a noticeable reduction in Energy MAE (typically around 12–16 meV). In contrast, further increasing $w$ from 10 to 15 or 20 brings only very minor improvements on the order of 1–2 meV in each size bin. Even for the largest molecules ($85 < N \leq 105$), the difference between using a moderate fixed window ($w = 10$) and larger windows is very small, and we do not observe any regime where large molecules require substantially larger $w$ to maintain accuracy. This suggests that, at the scale of the OC20 benchmark, a fixed moderate window size already captures the most important local geometric couplings for energy

prediction, and we find no empirical evidence that $w$ needs to grow explicitly with the total atom count $N$.

Table 18: Energy MAE vs. window size $w$ and molecule size (OC20 IS2RE, in-distribution validation; unit: meV).

| $w$ | $N \leq 45$ | $45 < N \leq 65$ | $65 < N \leq 85$ | $85 < N \leq 105$ |
|---|---|---|---|---|
| 5 | 378.6 | 381.1 | 384.6 | 380.3 |
| 10 | 363.5 | 371.2 | 376.5 | 367.1 |
| 15 | 362.1 | 368.8 | 375.1 | 366.0 |
| 20 | 361.6 | 368.9 | 374.7 | 365.4 |

From a modeling perspective, the tetrahedral interaction in TetraGT is designed to capture *local* geometric couplings within physically meaningful four-atom motifs. For any given bond angle or torsion angle, the most relevant interacting angles/torsions are typically confined to a limited chemical neighborhood (within a few bonds). The local coordination number of atoms is bounded by valence, which naturally imposes an upper bound on the number of physically meaningful neighboring angles/torsions. Consequently, a moderate window size (e.g., $w = 10$–$15$) is already sufficient in practice to cover the vast majority of such neighbors, while longer-range geometric dependencies can be propagated through stacked layers of angle and torsion tokens, rather than relying on a single attention layer with a very large $w$. Taken together, the empirical results and this geometric intuition suggest that TetraGT is robust to the choice of $w$ within a reasonable range; choosing a moderate $w$ effectively avoids local geometric under-coverage while striking a favorable trade-off between accuracy and computational efficiency.

