# OpenReview forum: "TetraGT: Tetrahedral Geometry-Driven Explicit Token Interactions with Graph Transformer for Molecular Representation Learning"
_ICLR.cc/2026/Conference — ICLR 2026 Poster_

### Official Review · Reviewer_uB9E · 2025-10-27

**Soundness:** 3
**Presentation:** 2
**Contribution:** 3
**Rating:** 6
**Confidence:** 3

**Summary:**

This work focuses on traditional molecular modeling, aiming to learn effective representations from 3D molecular configurations and then fine-tune them for specific downstream tasks (e.g., property prediction). The authors identify that previous methods often fail to incorporate higher-order geometric structures. To address this, they enhance Graph Transformer (GT) architectures by introducing bond angle and torsion angle information as tokens and designing a spatial tetrahedral attention mechanism. Additional techniques include a specially designed “directed cycle angle loss” and a hierarchical virtual node for molecule-level information aggregation. Experimental results demonstrate strong performance.

**Strengths:**

1. **Strong benchmark results.** The proposed method achieves excellent performance on several benchmarks (LIT-PCBA, PCQM4MV2, OC20 IS2RE, QM9), reaching or surpassing state-of-the-art results in nearly all cases. These results effectively demonstrate the transfer learning capability of the approach across diverse downstream tasks.
2. **Novel tetrahedral constraints.** The introduction of Tetrahedral Constraints and the Tetrahedral Interaction Module shows a novel perspective on modeling geometric relationships. The ablation study further confirms that this module contributes meaningfully to overall performance.
3. **Chirality structure analysis.** The authors conduct a statistical analysis of chirality structures (46.61% of the total 3M molecules). Table 9 compares errors with TGT, supporting the rationale behind their more complex design choices (e.g., attention mechanism and loss functions). The ablation study on the Directed Cycle Loss also validates the effectiveness of this component.
4. **Hierarchical virtual node.** The hierarchical virtual node design provides a certain degree of novelty and contributes to molecular-level information aggregation, though the innovation is not particularly substantial compared to other components.

**Weaknesses:**

1. **Efficiency concern.** As a work focused on model architecture design, the paper lacks analysis or discussion of model efficiency in the main text. Although Appendix E presents promising results, a more comprehensive discussion of efficiency in the main body would strengthen the paper’s overall contribution and practical relevance.
2. **Discussion of prior works on higher-order geometric structures.** Works such as [1,2] have also considered interactions among higher-order geometric structures, albeit in **implicit** ways (in contrast to the token-based approach proposed in this paper), as mentioned and discussed in Appendix J. Although the *“Implicit Modeling of Geometric Structures”* section in the Introduction provides some relevant statements, it would be helpful to clearly specify **which prior works** do not explicitly model higher-order geometric structures and **how** these structures are indirectly represented through combinations of atomic positions or pairwise relationships. Including this clarification in the main Introduction would better emphasize the motivation and novelty of the proposed approach.

[1] Wang Z, Liu G, Zhou Y, et al. Efficiently incorporating quintuple interactions into geometric deep learning force fields[J]. Advances in Neural Information Processing Systems, 2023, 36: 77043-77055.

[2] Wang Y, Wang T, Li S, et al. Enhancing geometric representations for molecules with equivariant vector-scalar interactive message passing[J]. Nature Communications, 2024, 15(1): 313.

**Questions:**

1. In Table. 12&13, the training and inference times of UniMol+ and TGT are missing. Since inference time is typically more important in practical applications, could you clarify why these results were omitted?

---

> ### Author Response · Authors · 2025-11-21
> **Response to Reviewer uB9E**
>
> Thank you for your positive assessment of our work. Below we address your questions and suggestions in detail.
>
> **Answer for W1:**
> Thank you for your suggestion. We have added a discussion of model efficiency in Section 4.4 of the Experiments (highlighted in blue). Due to the page limit of the main text, we are unable to fully elaborate on all details there, but we now explicitly direct readers who are particularly interested in efficiency to Appendix E. We appreciate your concern for the completeness of the manuscript.
>
> **Answer for W2:**
> Thank you for your suggestion. As requested, we have added a discussion of implicit geometric structure modeling in the third paragraph of the *Introduction* (highlighted in blue). We appreciate your attention to the rigor of the paper.
>
> **Answer for Questions:**
> We thank the reviewer for the careful and insightful comments. The missing training and inference time entries for some UniMol+ and TGT variants in Tables 12 and 13 were indeed an intentional omission on our side, made out of concern for the rigor of the reported results.
>
> Specifically, to ensure that the performance and efficiency comparisons are fair and reproducible, Tables 12 and 13 only report training and inference times for models that we have fully reproduced and benchmarked in our local environment. Previously, we performed systematic experiments and time measurements only for two large models, UniMol+ 77M and TGT 203M. Therefore, apart from the TetraGT series, the tables included training and inference times only for UniMol+ 77M and TGT 203M. The results for other UniMol+ and TGT variants were taken directly from their original papers, whose publicly available data do not provide corresponding training and inference time metrics; for this reason, these entries were uniformly omitted in the original manuscript.
>
> Fully recognizing your emphasis on inference efficiency in practical applications, in the revised version we have **supplemented the training and inference times for the other UniMol+ and TGT variants**, and have updated Tables 12 and 13 accordingly. The newly added content is highlighted in blue. We kindly invite you to review these updates, and we would be very grateful for any further comments or suggestions.

---

> > ### Comment · Reviewer_uB9E · 2025-11-21
> > **Response to authors' updates.**
> >
> > Thanks for the response and the new updates — my concerns have been resolved.
> >
> > - I will **raise my score to 8** due to the solid results and the comparatively better efficiency over UniMol+ and TGT.
> > - However, I will **keep my confidence at 3** because of the overall *heavy* design (as noted in *Reviewer bqeD, Weakness 4*). Although your response promises to release checkpoints for use, reproducing your work in a small-resource lab for **study** purposes would still be burdensome.

---

> > > ### Author Response · Authors · 2025-11-21
> > >
> > > We sincerely thank the reviewer for the increased score and the positive assessment of our results and efficiency.
> > >
> > > We fully understand your concern about the heavy overall design and the burden this may place on small‑resource labs. In the final release, we will provide not only pretrained checkpoints, but also complete training and fine‑tuning scripts, detailed configuration files, and smaller TetraGT variants that can be run on limited hardware for study and ablation purposes, without requiring full large‑scale pretraining. Our goal is to make the framework as accessible and reproducible as possible.

---

### Official Review · Reviewer_P24Z · 2025-10-29

**Soundness:** 3
**Presentation:** 2
**Contribution:** 3
**Rating:** 6
**Confidence:** 2

**Summary:**

TetraGT is a molecular representation model that explicitly incorporates 3D geometric parameters (bond angles and torsion angles) into graph transformer architectures. TetraGT treats angles and dihedrals as structured tokens and models their interactions based on tetrahedral geometry constraints. The method defines mathematical relationships among face and dihedral angles to ensure physically valid spatial configurations and introduces a multi-level attention mechanism that hierarchically updates representations from atoms to bonds, bond angles, and torsion angles. A specialized “tetrahedral interaction module” enables efficient communication among geometrically related triplets and quadruplets while reducing complexity via local sampling. Additionally, TetraGT introduces a directed circular angle loss to handle periodicity and chirality in angle prediction and employs hierarchical virtual nodes to integrate multi-level structural information for final molecular property prediction.

**Strengths:**

- Explicit “angle and torsion tokens” + “tetrahedral attention” are new combinations not seen in previous 3D graph Transformers.
- The physical constraints (Lemma 1) add rigor, not just architectural novelty.
- The directed cycle angle loss (DCA loss) for handling 2π-periodicity is conceptually sound and addresses chirality.
- The experiments show strong performance gains over baselines across multiple benchmarks, but clearer comparisons with recent equivariant models and ablations on key components would better substantiate the claimed improvements.

**Weaknesses:**

- While the model’s use of tetrahedral geometry constraints is mathematically grounded, the repeated use of the term “tetrahedral” may be confusing in a chemical context, since most molecular sites are not tetrahedral in bonding geometry. The paper should clarify that “tetrahedral” here refers to the geometric configuration of any four non-coplanar atoms (a 3D simplex), rather than chemically tetrahedral centers (sp³ atoms).

**Questions:**

There have been recent higher-order geometric message passing and equivariant Transformers (e.g., GemNet, Equiformer, TFN, SE(3)-Transformers, MACE) that already model angular and dihedral information. Does this explicit token-based formulation yield substantial conceptual or empirical advantage beyond existing geometric message passing frameworks?

---

> ### Author Response · Authors · 2025-11-20
> **Response to Reviewer P24Z- Part I**
>
> Thank you for the positive assessment of our work. Below we address your questions and suggestions in detail.
>
> **Answer for Weaknesses:**
> Thank you for your suggestion. We have added this clarification to the first paragraph of the *TetraGT architecture* subsection in the Method section (highlighted in blue). We appreciate your attention to the rigor of the manuscript.
>
> **Answer for Questions:**
> Thank you for your question. We address it in four parts below.
>
> - **Angle modeling in existing work**
>
>   Methods such as GemNet and related geometric GNNs typically expand distances, bond angles, and even dihedral angles into scalar features using radial and angular basis functions. These are then used as coefficients for triplet or quadruplet messages, which are passed along atoms or edges. In this way, angular information modulates the *strength* and *direction* of messages, but fundamentally still remains attached to atoms/edges, without independent angle nodes. Equivariant models such as Equiformer, TFN, SE(3)‑Transformer, and MACE, on the other hand, maintain higher‑order SE(3)‑equivariant tensor features for each atomic node. They use relative coordinates and spherical harmonics to construct equivariant convolution or attention kernels, where angles and many‑body geometry are implicitly encoded in the equivariant tensor field and realized through continuous kernels and tensor products, rather than modeling and supervising each specific bond angle or torsion as an explicit graph element.
>
> - **Implicit angle–angle relations and limitations of current Transformer designs**
>
>   In these frameworks, correlations *between angles* are mainly “mixed into” continuous kernel functions or higher‑order tensors; they exist as implicit high‑dimensional functional dependencies. As a result, they are difficult to explicitly constrain, visualize, or tune independently. Precise geometric inequalities and the exact relations among the three face angles and three dihedral angles within a tetrahedron can only be approximated indirectly through the overall loss and model capacity. This makes it easy to end up in situations where “each individual angle value is reasonable, but the joint combination within one tetrahedron is not physically consistent.”  Similarly, angular periodicity and chirality are mostly reflected indirectly through the coordinate field, rather than via losses designed directly in angle space. This often requires larger models and more data to learn stable angular patterns in chirality‑sensitive tasks and in 2D→3D geometric reconstruction. From the perspective of Transformer design, these models usually perform self‑attention only over a single token type, such as **atomic nodes** or **atom‑pair edges**. All orders of information (points, edges, triplets, quadruplets, higher‑order tensors) are repeatedly compressed into the same node/edge representation. This is effectively using a *single channel* to carry multi‑order structural and geometric relations, creating a clear information bottleneck: high‑order angle–angle dependencies must be implicitly encoded via complex nonlinearities within a single token’s limited dimensionality, making it hard to maintain a clear structural hierarchy and sufficient expressive power.

---

> ### Author Response · Authors · 2025-11-21
> **Response to Reviewer P24Z- Part II**
>
> - **Empirical advantages**
>
>   With this explicit design of angle tokens and their interactions, TetraGT demonstrates empirical advantages over traditional geometric GNNs and equivariant models on multiple tasks:
>
>   **Conformation prediction and geometric reconstruction:**  On distance, bond‑angle, and torsion prediction, TetraGT significantly outperforms variants without tetrahedral interaction or those that only use angles implicitly, across MAE, RMSE, and EwT metrics. The improvements are particularly pronounced for torsion‑angle errors and chirality prediction accuracy on large‑scale chiral molecules, indicating that joint multi‑angle constraints within local tetrahedra are better captured, rather than relying solely on energy‑level “error cancellation.”
>
>   **Large‑scale quantum chemistry and catalysis (PCQM4Mv2, OC20):**  Even under relatively weak geometric inputs (or in settings with only 2D graphs), TetraGT can match or surpass strong geometric baselines that rely on full 3D information. This shows that explicit angle tokens and their interactions indeed provide additional expressive power for alleviating information bottlenecks and reconstructing higher‑order geometry from topology.
>
>   **3D‑sensitive and 2D→3D‑dependent downstream tasks:**  On QM9, PDBBind, Peptides‑struct, and other tasks highly sensitive to fine‑grained 3D geometry and chirality, as well as on Peptides‑func and LIT‑PCBA where ground‑truth 3D conformations are unavailable and performance relies on a 2D→3D pretrained conformation predictor, the high‑quality geometric intermediate layer provided by angle tokens + tetrahedral interactions + DCA loss enables TetraGT to consistently outperform existing geometric GNNs / Transformers.
>
> - **Summary and positioning**
>
>   In summary, high‑order geometric message‑passing and equivariant Transformers represented by GemNet, Equiformer, TFN, SE(3)‑Transformer, and MACE have clearly demonstrated the importance of angular information. However, they primarily use angles as parameters of continuous kernels or messages, with angle–angle correlations implicitly encoded in high‑dimensional tensor fields that are carried by a single node/edge channel. This makes such correlations difficult to constrain and supervise explicitly and tends to create information bottlenecks. TetraGT, in contrast, **explicitly tokenizes bond angles and torsions**, builds multi‑order parallel structural channels, and applies dedicated geometric attention and periodic angle losses on tetrahedral subgraphs. This turns angle–angle relations from implicit, hard‑to‑control high‑dimensional functions into **graph‑level structures that are visible, constrainable, and optimizable in isolation**, while also mitigating the expressiveness bottleneck of packing all structural orders into a single token type. These design choices lead to tangible performance gains in conformation accuracy, chirality modeling, and 2D→3D pretraining and downstream transfer. Therefore, we view TetraGT as providing a **new and complementary geometric abstraction** on top of existing geometric message‑passing and equivariant modeling frameworks, rather than merely restating what current methods have already achieved.

---

### Official Review · Reviewer_nYFx · 2025-10-31

**Soundness:** 2
**Presentation:** 2
**Contribution:** 2
**Rating:** 4
**Confidence:** 4

**Summary:**

This paper proposes TetraGT, an attention-based model designed to accurately predict molecular properties from both 2D molecular graphs and 3D conformations. Existing methods often fail to directly model key molecular geometric parameters, such as bond angles and torsion angles, leading to limitations in capturing high-order structural relationships and local molecular chirality. TetraGT addresses these shortcomings through four core innovations: (1) The direct modeling of geometric parameters as structured tokens to prevent error accumulation from indirect atomic and bond representations; (2) A spatial tetrahedral attention mechanism, informed by tetrahedral geometry theory, to facilitate information exchange between these parameters; (3) An improved directed cycle angle loss function to handle geometric parameters and identify local chirality; (4) A hierarchical virtual node aggregation architecture that captures sub-structural information for a comprehensive molecular representation.
Experimental results demonstrate that TetraGT achieves superior overall performance on upstream datasets such as PCQM4Mv2 and OC20 IS2RE compared to models like Uni-Mol+. It also shows strong generalizability, outperforming models like EquiformerV2+NN on downstream tasks including QM9 and PDBBind.

**Strengths:**

1. The work is innovative in its direct representation of molecular geometry as structured tokens and the introduction of a spatial tetrahedral attention mechanism and a directed cycle angle loss function.
2. The model design is comprehensive and methodologically sound, effectively integrating geometric parameter interaction, local chirality identification, and graph sub-structure aggregation.
3. The model's performance and generalizability are rigorously validated across multiple upstream and downstream benchmark tasks.
4. This research provides a novel paradigm for molecular property prediction, with considerable potential for application in drug design and materials discovery.

**Weaknesses:**

1. The analysis of experimental results is insufficient. While Table 4 shows that TetraGT achieves state-of-the-art performance on 5 out of 12 metrics on the QM9 dataset, it is outperformed on the remaining 7 metrics by models like EquiformerV2+NN. The manuscript lacks a systematic discussion to explain this performance disparity.
2. The organization of the related work section is non-standard. Placing the "Related Work" section after the "Experiments" and immediately before the "Conclusion" deviates from conventional academic structure.
3. The introduction is poorly articulated. The summary of key contributions in the "Introduction" section is not itemized, which hinders readability and fails to clearly preview the paper's innovations.

**Questions:**

1. A systematic explanation should be provided in the main text to account for the performance gap on the specific QM9 metrics where TetraGT does not achieve optimal results.
2. The "Related Work" section should be relocated to follow the "Introduction" and precede the "Method" section. Content in the introduction that details core challenges and innovations (e.g., the rationale for the spatial tetrahedral attention mechanism) should be integrated into this restructured "Related Work" section to better contextualize the research.
3.  The key contributions within the "Introduction" section should be presented as a clear, itemized list to enhance readability and provide a straightforward overview of the paper's novel elements.

---

> ### Author Response · Authors · 2025-11-20
> **Response to Reviewer nYFx**
>
> Thank you for the reviewer’s questions and suggestions. Since the contents of *Weakness* and *Questions* are largely the same, we directly respond to the items listed under *Questions* below.
>
> **Answer for Q1:**
>
> As shown in Table 4, TetraGT achieves current state‑of‑the‑art (SOTA) results on 5 out of 12 molecular properties, with particularly notable improvements on energy‑related quantities such as the HOMO level $\epsilon_H$, LUMO level $\epsilon_L$, energy gap $\Delta\epsilon$, and constant‑volume heat capacity $C_v$. Among the remaining 7 properties, TetraGT attains the **second‑best** result on 2 of them, and for the others the numerical gaps to the best method are generally small. Overall, TetraGT is competitive among current methods. Therefore, our QM9 results can be summarized as follows: TetraGT exhibits clear advantages for a class of properties that are strongly tied to energy and electronic structure, while maintaining overall competitiveness on the other properties, but it does not uniformly surpass all models such as EquiformerV2+NN on all 12 metrics.
>
> The core reason for this difference lies in the **degree of physical alignment between the pretraining objectives and the downstream properties**. As discussed in Appendix G.1, the pretraining task of TetraGT on PCQM4Mv2 is to predict the HOMO–LUMO gap and to denoise molecular conformations in terms of distances and angles. This primarily strengthens the model’s capability to capture the **consistency between global electronic structure and 3D geometry**. Consequently, in QM9, those properties that are physically very close to the HOMO–LUMO gap (such as $\epsilon_H$, $\epsilon_L$, $\Delta\epsilon$, and several other energy‑related quantities) can benefit directly, leading to the most significant improvements for TetraGT on these metrics. In contrast, for properties that depend more strongly on long‑range charge distribution, polarization, and overall spatial extent (e.g., $\mu$, $\alpha$, $\text{ZPVE}$, $R^2$), our current pretraining objective is not the best possible match. As a result, TetraGT remains very close to SOTA performance on these tasks, but does not surpass highly specialized models on all 12 properties.
>
> From the perspective of model design, EquiformerV2+NN is a specialized SE(3)‑equivariant model with a strong inductive bias for continuous 3D coordinates and their derivatives. This gives it advantages on certain QM9 metrics that focus on finely fitting energies. In contrast, the goal of TetraGT is to provide a **general geometric and chiral representation framework** that can transfer across different systems and tasks. It explicitly introduces bond‑angle/torsion tokens and tetrahedral geometric interactions to enhance local conformation and chirality modeling, and achieves unified SOTA or highly competitive results on large‑scale quantum chemistry (PCQM4Mv2), catalysis (OC20 IS2RE), protein–ligand binding (PDBBind), peptide structure and function (Peptides‑struct / Peptides‑func), as well as small‑molecule tasks with only 2D information (LIT‑PCBA). To maintain this generality, we adopt a unified pretraining objective and model architecture across datasets, rather than performing task‑specific optimization for each individual QM9 property. This “generalization vs. task specialization” trade‑off is explicitly discussed in Appendices G.1 and H.
>
> In response to the reviewer’s concerns, **we have added a dedicated subsection in Appendix G.1** to systematically analyze the performance differences across different QM9 properties, focusing on (i) how well the pretraining objectives match each property type, and (ii) the differing design philosophies between equivariant energy models and our angle‑driven representation framework. **We have also added a summary paragraph in the main text (Section 4.2)** to explain why TetraGT shows more pronounced advantages on orbital/energy‑related properties and complex downstream tasks, while some highly specialized models may still hold a slight edge on certain QM9 properties. We believe these additions more clearly convey TetraGT’s overall strengths, its performance boundaries, and its intended application scenarios. All new content is highlighted in blue.
>
> **Answer for Q2 and Q3:**
>
> Following your advice, we have reorganized and adjusted the paper as follows: the Related Work section has been moved forward to appear immediately after the Introduction and before the Method section. We also integrated and rewrote parts of the Introduction and Related Work that describe the core challenges and key contributions to avoid redundancy and improve clarity. We have revised the Introduction to present the main contributions in a clear, itemized list. These changes are highlighted in blue in the latter half of the Introduction and in the Related Work section of the revised manuscript. We kindly invite you to review these modifications and would greatly appreciate any further guidance and suggestions.

---

### Official Review · Reviewer_bqeD · 2025-11-01

**Soundness:** 2
**Presentation:** 2
**Contribution:** 3
**Rating:** 4
**Confidence:** 4

**Summary:**

This paper proposes TetraGT, a graph Transformer framework for molecular representation learning. The key idea is to treat not only atoms and chemical bonds as tokens, but also higher-order geometric structures, specifically bond angles and torsion angles, as explicit tokens that can directly interact through an attention mechanism constrained by tetrahedral geometric consistency, rather than relying on traditional models to infer such information implicitly from pairwise distances. The paper further introduces a chirality-aware angular learning objective, the Directed Cycle Angle Loss, which models angles as directed periodic variables over (0,2π); this is intended to distinguish local chirality and avoid discontinuities around cases that occur in standard angle regression or classification. In addition, the model employs a hierarchical virtual node design that aggregates information separately at the atom, bond, angle, and torsion levels before passing it to a global node, aiming to alleviate the information bottleneck of a single global aggregator. The authors argue that these components together enable more faithful modeling of stereochemistry, local chirality, and conformational stability, thereby improving representation quality and transferability.

**Strengths:**

1. The paper directly addresses three known gaps: explicit modeling of chirality, explicit treatment of higher-order geometry (bond and torsion angles), and explicit handling of geometric consistency constraints.

2. The proposed tetrahedral interaction module constrains attention among angle/torsion tokens to local tetrahedral units and encodes geometric consistency, aiming to retain physical validity while avoiding naive O(N^3)high-order connectivity.

3. The Directed Cycle Angle Loss treats angles as directed periodic variables over (0,2π), which is intended to distinguish local chirality and avoid instability at angular wrap-around. Most prior models do not handle it directly.

4. The model promotes atoms, bonds, bond angles, and torsions to first-class tokens and aggregates them via hierarchical virtual nodes, rather than relying on a single global node. This is meant to reduce information bottlenecks.

5. The method is evaluated on diverse and competitive benchmarks (PCQM4Mv2, OC20 IS2RE, QM9, PDBBind, Peptides, LIT-PCBA) and reports state-of-the-art or near state-of-the-art results against strong baselines, suggesting transferability beyond a single task

**Weaknesses:**

1. The method treats local groups of atoms as “tetrahedral units” and uses these units as the fundamental template for constrained attention and geometric consistency. However, the paper does not demonstrate how well this assumption holds in other common chemical settings (e.g., conjugated rings, metal coordination sites) where local geometry is not tetrahedral. It is unclear whether this inductive bias could introduce systematic errors in such non-tetrahedral regimes, or whether the model adapts automatically?

2. The tetrahedral geometric inequalities are injected as attention biases/gates, and the authors argue this both reduces complexity from 𝑂(𝑁3) to approximately 𝑂(𝑤𝑁2) and enforces physical consistency. However, the paper does not provide a formal characterization of this mechanism. In particular, it does not report metrics such as the fraction of geometrically invalid local angle/torsion configurations before vs. after training, nor does it prove that predicted angles/torsions are always physically realizable. It remains unclear whether this acts as a true constraint or mainly as an inductive bias.

3. The method relies on a local sampling window of size 𝑤 to decide which angle/torsion tokens are allowed to interact. If 𝑤 fails to cover an important substructure, or only partially covers it, the model could miss critical geometric couplings. The paper does not present sensitivity studies on different 𝑤 values or analyze the impact of under-coverage on accuracy, nor does it clarify how 𝑤 should scale with molecular size.

4. Training still requires on the order of tens of A100 GPU-days, comparable to other top-performing large models, which is a high barrier for many labs. The authors assert that the approach is scalable to larger systems, but they do not analyze how resource usage grows when moving to substantially larger molecules or protein–ligand complexes. In particular, it is not clear whether the number of tokens and pairwise interactions will scale roughly linearly or blow up faster, so the scalability claim is not yet quantitatively supported.

5. In the abstract, Figure 1, and the conclusion, the authors use the alternate name “TDGT” for the proposed architecture but do not provide a proper definition.

**Questions:**

1. The method treats local groups of atoms as “tetrahedral units” and uses these units as the fundamental template for constrained attention and geometric consistency. However, the paper does not demonstrate how well this assumption holds in other common chemical settings (e.g., conjugated rings, metal coordination sites) where local geometry is not tetrahedral. It is unclear whether this inductive bias could introduce systematic errors in such non-tetrahedral regimes, or whether the model adapts automatically?

2. The tetrahedral geometric inequalities are injected as attention biases/gates, and the authors argue this both reduces complexity from 𝑂(𝑁3) to approximately 𝑂(𝑤𝑁2) and enforces physical consistency. However, the paper does not provide a formal characterization of this mechanism. In particular, it does not report metrics such as the fraction of geometrically invalid local angle/torsion configurations before vs. after training, nor does it prove that predicted angles/torsions are always physically realizable. It remains unclear whether this acts as a true constraint or mainly as an inductive bias.

3. The method relies on a local sampling window of size 𝑤 to decide which angle/torsion tokens are allowed to interact. If 𝑤 fails to cover an important substructure, or only partially covers it, the model could miss critical geometric couplings. The paper does not present sensitivity studies on different 𝑤 values or analyze the impact of under-coverage on accuracy, nor does it clarify how 𝑤 should scale with molecular size.

4. Training still requires on the order of tens of A100 GPU-days, comparable to other top-performing large models, which is a high barrier for many labs. The authors assert that the approach is scalable to larger systems, but they do not analyze how resource usage grows when moving to substantially larger molecules or protein–ligand complexes. In particular, it is not clear whether the number of tokens and pairwise interactions will scale roughly linearly or blow up faster, so the scalability claim is not yet quantitatively supported.

5. With respect to the overall pipeline: in the multi-stage training process, could you clarify the relationship between the “conformation predictor” and the “task predictor”? It would also be helpful to more explicitly describe the full loss composition across stages — is it primarily pairwise atomic distance regression together with the Directed Cycle Angle Loss for angles, or are there additional property-prediction heads contributing to the objective?

6. (Minor question) In the abstract, Figure 1, and the conclusion, the authors use the alternate name “TDGT” for the proposed architecture but do not provide a proper definition.

---

> ### Author Response · Authors · 2025-11-20
> **Response to Reviewer bqeD- Part I**
>
> Thank you for your questions and suggestions. Since the contents of *Weakness* and *Questions* are the same, we directly respond to the items listed under *Questions* below.
>
> **Answer for Q1:** We appreciate the reviewer’s question regarding the applicability of the “tetrahedral unit” in non‑typical local geometries such as conjugated rings and metal coordination environments. This is indeed a very important point and has helped us reflect on and further clarify the assumptions and geometric meaning of Lemma 1. Our response can be summarized in three main points:
>
> - **The tetrahedral unit is a general four‑point geometric template; its limiting behavior in conjugated/coordination scenarios can be interpreted as a degenerate case, which does not cause issues for realistic 3D conformations, and the DCA loss is designed to handle boundary angles such as 0/180°.**
>   In our paper, the “tetrahedral unit” simply refers to the local geometric relationship among any four atoms in three‑dimensional Euclidean space. Lemma 1 uses general geometric consistency constraints for four points that can be embedded in 3D space (triangle relations and inequalities between face angles and dihedral angles), without assuming any specific ideal configuration.
>
>   For the situations explicitly mentioned by the reviewer: in an ideal conjugated planar ring or an ideal square‑planar coordination environment, if the four selected atoms are **strictly coplanar**, then these four points form a geometrically “degenerate tetrahedron” with zero volume. In this limiting case, the “strict inequalities” in Lemma 1(a)(b) degenerate into equalities, and in (c) the denominator approaches 0 when some face angles are 0 or $\pi$.
>
>   However, in realistic molecular geometries or DFT‑optimized structures, these “ideal planes” almost always exhibit slight warping, i.e., the four atoms are numerically not exactly coplanar but form a “nearly planar yet non‑zero volume” small tetrahedron. In this situation, the four points still constitute a genuine tetrahedron, and the inequalities and angle–dihedral relations in Lemma 1 remain mathematically valid in a strict sense. Therefore, using these geometric relations in real 3D conformations (including nearly planar conjugated/coordination systems) is both reasonable and stable.
>
>   In addition, for the many torsion angles close to 0/$\pi$/2$\pi$ that frequently occur in conjugated rings, aromatic systems, and many coordination environments, we specifically designed the Directed Cycle Angle Loss (DCA), which treats angles as periodic variables and aligns them on the angular circle. This avoids the discontinuous error of traditional linear losses at the 0/2$\pi$ boundary, while preserving direction information to capture chirality. In other words, these “nearly planar” geometries are not only covered by our geometric template, but are also specifically and carefully handled in our loss design.
>
>   We also appreciate the reviewer’s reminder: in the revised version, we have made the assumptions of Lemma 1 more explicit so that the mathematical statements are more rigorous. The modifications are highlighted in blue in Lemma 1.
>
> - **The tetrahedral angle constraints act as soft constraints that restrict communication to “neighboring angle tokens” rather than enforcing any rigid geometric shape.**
>   Our use of tetrahedral geometry does not “hard‑project” structures into a fixed shape. Instead, in the spirit of the triangle inequality used in TGT and Uni‑Mol+, it provides a reasonable communication structure for angle tokens. In TGT / Uni‑Mol+, the triangle inequality leverages triangle relations so that edge tokens sharing a node can communicate directly without going through that node; and because only two edges that share a node can form a valid triangle, this constraint naturally restricts interactions to “adjacent edges” rather than arbitrary edge pairs.
>
>   Similarly, in TetraGT, we only establish higher‑order interactions between **neighboring angle tokens** that belong to the same four‑atom unit substructure (i.e., face angles/dihedral angles sharing that four‑atom unit), and we do not allow indiscriminate communication between arbitrary pairs of angles. Since angles that are geometrically far apart typically lack direct chemical relevance, this design of “interaction only between locally adjacent angles” aligns with chemical intuition.
>
>   In implementation, the tetrahedral geometry only influences attention via soft mechanisms such as biases and gating functions (e.g., $b_f(\cdot)$, $g_f(\cdot)$), which gently modulate the attention weights and “guide angle tokens toward appropriate communication partners.” It does not forcibly deform local structures into any specific geometry, thereby reducing the risk of introducing systematic bias under non‑typical configurations.

---

> ### Author Response · Authors · 2025-11-20
> **Response to Reviewer bqeD- Part II**
>
> - **On real datasets containing many conjugated rings and coordination structures, we do not observe systematic degradation; instead, we see overall performance improvements.**
>   Empirically, the benchmark datasets we use contain many non‑typical local geometries that the reviewer is concerned about. PCQM4Mv2, QM9, LIT‑PCBA, and the ligands in PDBBind all contain a large number of conjugated rings and aromatic systems; OC20 and some PDBBind complexes involve diverse metal coordination and metal surface environments.
>   On these tasks, TetraGT consistently outperforms strong baselines such as TGT and Uni‑Mol+ in terms of HOMO‑LUMO gap prediction, binding affinity prediction, structure regression, and virtual screening metrics. We do not observe any phenomenon where performance degrades on tasks with a higher proportion of conjugated/coordination structures. These results indicate that the introduced tetrahedral inductive bias does not lead to observable systematic bias in “non‑tetrahedral” scenarios such as conjugated rings and metal coordination, but instead effectively enhances the modeling of angle‑dependent geometric relationships.
>
> **Answer for Q2:** Thank you for your question regarding the tetrahedral attention mechanism. Our response is as follows：
>
> - **Form of the constraint**
>   In TetraGT, the tetrahedral geometric inequalities enter the model through the attention bias and gate (Equations (5)/(6)); that is, local angle/torsion combinations that satisfy Lemma 1 receive larger attention weights, while combinations that clearly violate the geometric inequalities are suppressed by the gate. Therefore, in implementation it is a data‑driven **soft constraint / strong inductive bias**, rather than a hard constraint that explicitly “projects” the output angles.
> - **Complexity characterization**
>   As shown in Appendix E.1, for a molecule with $N$ atoms, the numbers of bond angles and torsion angles are both $O(N)$. If we were to perform full higher‑order attention over all triplets/quadruplets, the complexity would reach $O(N^3)$ or even $O(N^4)$. TetraGT only establishes higher‑order interactions within local tetrahedra formed by shared vertices/shared faces and, for each structural token, retains only $w$ nearest neighbors. Consequently, the higher‑order part has complexity $O(wN^2)$, the backbone atom–bond–angle–torsion attention also has complexity $O(wN^2)$, and the overall complexity is $O(wN^2)$. Appendix E.1 provides a detailed derivation of this result, as well as a time‑complexity comparison with Uni‑Mol+ and TGT.
>
> - **Empirical characterization of physical consistency**
>   Following the reviewer’s suggestion, we conducted additional experiments and statistics on PCQM4Mv2. Using the same setup as in Appendix D, we compared TetraGT **with** and **without** the tetrahedral mechanism, and computed the fraction of local structures in the predicted conformations that violate Lemma 1(a)(b)(c). (For Lemma 1(c), a configuration is considered “invalid” if its numerical error exceeds 10%.) The results are summarized below:
>
>   | Quantity                                                     | TetraGT (w/o tetrahedral mechanism) | TetraGT |
>   | ------------------------------------------------------------ | ----------------------------------- | ------- |
>   | Fraction violating bond-angle inequalities (Lemma 1(a)) (%)  | 6.31                               | 1.26   |
>   | Fraction violating bond-angle inequalities (Lemma 1(b)) (%)  | 8.45                               | 2.17   |
>   | Fraction violating bond-angle inequalities (Lemma 1(c)) (%)  | 13.60                              | 2.68   |
>
>   We can see that, after introducing the tetrahedral mechanism, the fraction of local structures violating the geometric inequalities is reduced by about 3–5×, indicating that this mechanism **significantly compresses the probability mass of geometrically unrealizable regions under the training distribution**.
>
>   Moreover, Appendices C and D further characterize consistency with DFT reference geometries from a more application‑oriented perspective: on 1.77 million molecules containing chiral centers, compared with TGT, TetraGT reduces the torsion MAE from 0.597 to 0.312 rad and improves local chirality prediction accuracy from 32.5% to 76.7%. On validation‑3D, the continuous distance/angle errors and EwT metrics of conformations reconstructed by the refiner are also consistently better than those of TGT.
>
>   Taken together, these results show that, although the tetrahedral geometry is implemented as a soft constraint in the model, it nonetheless strongly guides the model, in a statistical sense, towards physically realizable local geometric configurations.

---

> ### Author Response · Authors · 2025-11-20
> **Response to Reviewer bqeD- Part III**
>
> **Answer for Q3:** We thank the reviewer for pointing out that the local sampling window $w$ was not analyzed in detail. We have conducted additional experiments and provide the following analysis:
>
> - We systematically examined the impact of different $w$ values on conformation prediction and energy prediction on the in-distribution validation set of OC20 IS2RE, setting $w \in \{5, 10, 15, 20\}$, and evaluated distance MAE, bond-angle MAE, torsion-angle MAE, Energy MAE, as well as the relative training time:
>
>   | $w$  | Distance MAE (Å) ↓ | Bond-angle MAE (rad) ↓ | Torsion-angle MAE (rad) ↓ | Energy MAE (meV) ↓ | Training time / epoch (relative, 1.0 = default) |
>   | ---- | ------------------ | ---------------------- | -------------------------- | ------------------ | ----------------------------------------------- |
>   | 5    | 0.191              | 0.297                  | 0.387                      | 382.4              | 0.76                                            |
>   | 10   | 0.168              | 0.228                  | 0.316                      | 373.2              | 1.0                                             |
>   | 15   | 0.163              | 0.221                  | 0.313                      | 372.6              | 1.52                                            |
>   | 20   | 0.157              | 0.217                  | 0.309                      | 372.1              | 2.83                                            |
>
>   The experiments show that when $w$ increases from 5 to 10, both geometric and energy predictions improve noticeably. For example, Energy MAE decreases from 382.4 meV to 373.2 meV, and the bond-angle/torsion-angle MAEs also drop significantly. However, when $w \ge 10$, further increasing $w$ from 10 to 15 or 20 yields only very small improvements in Energy MAE (about 0.6–1.1 meV), while the training time per epoch grows from 1.0× to 1.52× and 2.83×. Overall, this exhibits a trend of “rapidly saturating performance gains versus continuously increasing computational cost”. This indicates that once $w$ exceeds a moderate threshold, TetraGT becomes quite robust to the choice of $w$, and we do not observe obvious accuracy degradation due to insufficient local coverage. A medium-sized $w$ thus provides a reasonable trade-off between accuracy and efficiency.
>
> - To investigate whether $w$ needs to scale with molecular size, we further divided the OC20 IS2RE ID validation set into four ranges according to the number of atoms, and computed Energy MAE for different $w$ values within each range:
>
>   | $w$ / Atom\_num (N) | $N \le 45$ Energy MAE ↓  | $45 < N \le 65$ Energy MAE ↓  | $65 < N \le 85$ Energy MAE ↓  | $85 < N \le 105$ Energy MAE ↓  |
>   | ------------------- | --------------------- | -------------------------- | -------------------------- | --------------------------- |
>   | 5                   | 378.6                 | 381.1                      | 384.6                      | 380.3                       |
>   | 10                  | 363.5                 | 371.2                      | 376.5                      | 367.1                       |
>   | 15                  | 362.1                 | 368.8                      | 375.1                      | 366.0                       |
>   | 20                  | 361.6                 | 368.9                      | 374.7                      | 365.4                       |
>
>   The results show that, in all atom-count ranges, increasing $w$ from 5 to 10 consistently reduces the error by about 12–16 meV, whereas further increasing $w$ from 10 to 15 or 20 only brings marginal improvements of about 1–2 meV in each range. Even in the largest atom-count range $85 < N \le 105$, using a fixed medium window (e.g., $w = 10$) versus a larger window leads to only very minor performance differences. We do not observe any phenomenon where “large molecules must rely on a larger $w$ to maintain accuracy”. Therefore, at the current benchmark scale, a fixed medium $w$ is already sufficient to cover the most critical geometric couplings for the prediction tasks, and it is not necessary to explicitly scale $w$ with molecular size.
>
> - From a method-design perspective, the tetrahedral interactions in TetraGT are intended to capture geometric couplings within local four-atom configurations. For any given bond angle or torsion angle, the most physically relevant angles/torsions are primarily concentrated within a limited chemical distance, and the local atomic coordination number is bounded by valence constraints. As a result, a medium-sized window (e.g., $w = 10 \sim 15$) is already sufficient to cover the vast majority of physically meaningful neighboring angles/torsions. Longer-range geometric dependencies can be propagated layer by layer through stacked structural tokens, rather than relying on extremely large $w$ within a single layer.

---

> ### Author Response · Authors · 2025-11-20
> **Response to Reviewer bqeD- Part IV**
>
> Combining the empirical results above with this molecular-geometric intuition, we consider setting $w = 10$ in TetraGT to be a good choice that balances computational efficiency and predictive accuracy, and this window size exhibits good robustness across different molecular sizes. We have added the corresponding experiments and analyses on the choice of $w$ and its relationship with molecular size to the revised manuscript in Appendix K “Sensitivity Analysis of the Window Size $w$”. We respectfully invite the reviewer to review this section, and again sincerely thank you for your valuable comments.
>
> **Answer for Q4:**
> We thank the reviewer for the question regarding scalability. Large‑scale pretraining indeed requires tens of days of A100 GPU time, which is on the same order of magnitude as other recent high‑performance 3D graph Transformers (such as Uni‑Mol+ and TGT). We fully agree that it is not reasonable to expect every lab to repeat this large‑scale pretraining process. Our plan is to **publicly release the pretrained model weights**, so that most users only need to perform lightweight downstream fine‑tuning.
>
> To **quantitatively** support our claim that the method “scales with system size,” we added a scaling analysis on PDBBind v2016. We partition the systems into multiple ranges by size, and in each range we compute: the average number of atoms `mean_atoms`, the average number of bond angles `mean_bond_angles`, the average number of torsions `mean_torsions`, and the **peak memory usage per training step** (normalized relatively). The statistics are shown in the table below:
>
> | Metric           | (0.0, 20.0] | (20.0, 40.0] | (40.0, 60.0] | (60.0, 80.0] | (80.0, 100.0] | (100.0, 120.0] | (120.0, 140.0] | (140.0, 160.0] | (160.0, 180.0] |
> | ---------------- | ----------- | ------------ | ------------ | ------------ | ------------- | -------------- | -------------- | -------------- | -------------- |
> | mean_atoms       | 14.64       | 28.97        | 47.74        | 70.15        | 90.06         | 108.90         | 127.41         | 148.07         | 166.89         |
> | mean_bond_angles | 20.83       | 44.76        | 70.85        | 99.07        | 126.61        | 151.62         | 174.77         | 204.18         | 233.67         |
> | mean_torsions    | 24.98       | 57.58        | 88.15        | 119.16       | 151.02        | 179.21         | 204.83         | 240.25         | 275.44         |
> | 平均相对峰值显存 | 1.0         | 2.24         | 6.79         | 19.87        | 33.16         | 61.60          | 97.31          | 130.68         | 165.39         |
>
> We observe that when the average number of atoms increases from 14.6 to 166.9 (about 11×), the number of bond‑angle tokens increases from 20.8 to 233.7, and the number of torsion tokens from 25.0 to 275.4. Across all ranges, `mean_bond_angles` and `mean_torsions` remain approximately linear in `mean_atoms`, with a largely stable ratio of about 1.4–1.7 angle/torsion tokens per atom. This is consistent with the theoretical analysis: in realistic molecular and protein–ligand systems, atomic valence is bounded, so
> $|E| = O(N),\quad |B| = \sum_i \binom{\deg(i)}{2} = O(N),\quad |T| = O(N),$
>
> i.e., **the numbers of bond‑angle and torsion tokens are $O(N)$ on real data**, and there is no quadratic or higher‑order “combinatorial explosion” with respect to $N$.
>
> We also measure the **peak memory usage per training step** in each range. As the average number of atoms increases by about 11× (14.6 → 166.9), the normalized peak memory increases from 1.0 to about 165.4×. Fitting a power law to the relationship between “peak memory” and “number of atoms” yields an exponent close to 2, which is fully consistent with the $O(N^2)$ complexity of standard full‑graph attention. This result indicates that the newly introduced tetrahedral geometric interaction module **does not introduce additional $O(N^3)$ or $O(N^4)$ blow‑ups in practice**, but instead maintains the same $O(N^2)$ order as other graph Transformers, adding only a limited constant‑factor overhead.

---

> ### Author Response · Authors · 2025-11-20
> **Response to Reviewer bqeD- Part V**
>
> These empirical scaling results are consistent with our complexity and runtime comparisons in **Appendix E**. In Appendix E, under the same depth and width settings, we compare TetraGT with representative advanced models such as Graphormer, Uni‑Mol+, and TGT, and we find that:
>
> 1. **Theoretical time and memory complexity**: The complexity of Uni‑Mol+ and TGT is $O(N^3)$, whereas that of TetraGT is $O(wN^2)$, where the local sampling window $w$ is reasonably designed. This is precisely what enables TetraGT to scale to larger molecular datasets such as PDBBind and Peptides‑struct.
> 2. **Actual training and inference time**: Under the same hardware and batch settings, the per‑step runtime of TetraGT is very close to that of these strong baselines, and in some configurations even slightly lower.
> 3. In Appendix E, we also report smaller TetraGT variants which, despite substantially reduced computational budgets (in terms of parameter count and training steps), still achieve competitive or even superior performance to existing methods on multiple tasks.
>
> Finally, we would like to reiterate that **large‑scale pretraining is a one‑time cost**, similar to most current large‑model approaches. Upon official release, we will provide the pretrained TetraGT weights. Therefore, labs with limited computational resources will not need to repeat tens of days of pretraining from scratch; instead, they can reproduce and leverage our model’s capabilities by performing relatively short‑term fine‑tuning on downstream datasets (typically feasible on a single GPU or a small number of GPUs).
>
> **Answer for Q5:**
> We thank the reviewer for the comment and will explain the overall pipeline in detail:
>
> In our framework, we use two TetraGT instances that share the same architecture but have independent parameters: a **conformation predictor** and a **task predictor**. The conformation predictor is trained only in the first stage, where it predicts 3D geometry (pairwise distances, bond angles, and torsions) from 2D molecular graphs. After this stage, it is fully frozen in all subsequent stages. The task predictor is trained in the second and third stages, where it takes noisy or predicted geometric features as input to perform specific property prediction tasks, and simultaneously learns from auxiliary geometric denoising targets to better model geometric information.
>
> For downstream tasks that provide only 2D but no ground‑truth 3D, we first use the frozen conformation predictor to generate geometric features from the 2D graph, and then feed them into the task predictor for property prediction. For tasks with ground‑truth 3D structures, the task predictor directly uses the true geometry.
>
> In the **first stage (conformation prediction)**, we train only the conformation predictor. It outputs discrete bins for all pairwise distances, bond angles, and torsions. The loss is the sum of a cross‑entropy loss over pairwise distances and the directed cycle angle loss (DCA) over angles (applied to both bond angles and torsions), written as
>
> $$
> L_{\text{conf}}=L_{\text{dist-CE}} + \alpha L_{\text{angle-DCA}} ,
> $$
>
> where $L_{\text{dist-CE}}$ is the cross‑entropy loss over discretized distances, and $L_{\text{angle-DCA}}$ is our proposed DCA loss. The coefficient $\alpha$ controls the relative weight between distance and angle losses; in our experiments, we use a distance:angle ratio of about 1:4. This stage does not include any property prediction or other self‑supervised heads.
>
> In the **second stage (pretraining on PCQM4Mv2 / OC20)**, we train only the task predictor. The inputs are distances and angles from perturbed (noisy) ground‑truth conformations. The outputs are, on the one hand, the main task properties (e.g., HOMO–LUMO gap or adsorption energy), and on the other hand, denoised predictions of distances and angles. The total loss is
>
> $$
> L =L_{\text{task}} + \lambda_{\text{conf}} \big( L_{\text{dist-CE}} + L_{\text{angle-DCA}} \big) ,
> $$
>
> where $L_{\text{task}}$ is the regression loss for the main property prediction task, and the geometric term in parentheses is again the combination of pairwise‑distance cross‑entropy and angle DCA loss, now serving as an auxiliary “noisy‑geometry denoising” objective weighted by $\lambda_{\text{conf}}$. In the paper, we do not use any other self‑supervised or auxiliary attribute‑prediction heads beyond these two types of geometric heads.

---

> ### Author Response · Authors · 2025-11-20
> **Response to Reviewer bqeD- Part VI**
>
> In the **third stage (downstream fine‑tuning)**, we continue training the task predictor, with two different cases:
>
> - If the downstream task has ground‑truth 3D conformations (e.g., QM9, PDBBind, Peptides‑struct), we directly use the true geometry (optionally with small noise) as input, and the target is the downstream property. We may optionally keep the geometric denoising auxiliary objective, with total loss
>   $ L=L_{\text{task,downstream}} + \lambda_{\text{conf}}^{\text{FT}} \big( L_{\text{dist-CE}} + L_{\text{angle-DCA}} \big) , $
>
> - If the downstream task has only 2D molecular graphs but no ground‑truth 3D (e.g., Peptides‑func, LIT‑PCBA, the validation set of PCQM4Mv2), we first use the frozen conformation predictor to generate predicted distances and angles from 2D, and then feed them into the task predictor. Since no ground‑truth geometry is available, we only use the downstream property loss:
>   $ L=\mathcal{L}_{\text{task,downstream}}.$
>
>   No geometric reconstruction loss is included in this case.
>
> In summary, all objectives across the multi‑stage training can be described as follows:
> - The first stage uses only geometric reconstruction losses (pairwise‑distance cross‑entropy and angle DCA).
> - The second and third stages use a combination of “main property‑task loss + weighted geometric denoising loss (enabled only when 3D labels are available)”.
>
> Besides these two geometric prediction heads and the property prediction head, we do not introduce any additional heads or self‑supervised tasks.
>
> ---
>
> **Answer for Minor Question:**
> Thank you for pointing out the abbreviation error in the name. We have corrected this in the revised version.

---

### Meta-Review · Area_Chair_pggV · 2026-01-08

**Summary:**

This paper proposes TetraGT, a graph transformer architecture that treats bond angles and torsion angles as explicit tokens , using a "tetrahedral" attention mechanism to enforce geometric consistency. The initial reviews were borderline and mixed (two "4:Marginally Below, two "6: Marginally Above"), but sentiment improved during the rebuttal, with one reviewer raising their score to "8:Accept". While reviewers praised the novel explicit geometric tokenization and strong results on PCQM4Mv2/OC20, major concerns focused on the validity of the "tetrahedral" assumption for non-tetrahedral chemical groups (e.g., planar rings), the sensitivity to the local window size parameter, and the computational scalability of the token-heavy approach.

**Reviewer Concerns:**

**Addressed**:
- Tetrahedral Assumption (bqeD, P24Z): Reviewers worried the "tetrahedral" bias would fail for planar/conjugated systems. The authors provided mathematical clarification that any 4 points form a geometric tetrahedron (even if degenerate) and showed empirical results on ring-heavy datasets demonstrating no performance degradation.
- Window Size Sensitivity (bqeD): bqeD requested sensitivity analysis on the window size $w$. The authors provided new experiments showing performance saturates at $w=10$, proving the method is robust to this hyperparameter.
- Efficiency Benchmarks (uB9E): Missing inference/training time comparisons were added to the tables as requested.
- Constraint Mechanism (bqeD): Clarified that the geometric constraints are soft inductive biases. The authors provided new statistics showing a 3-5x reduction in geometric violations after training.
- Writing (P24Z,  nYFx): Clarified that "tetrahedral" refers to the geometric simplex of 4 atoms, not chemical hybridization. Reorganized paper as recommended by nYFx.

**Partially Addressed**:
- QM9 Performance Gaps (nYFx): The reviewer noted TetraGT loses to EquiformerV2 on 7 out of 12 metrics. The authors explained this as a trade-off: their pretraining objective (HOMO-LUMO) aligns with energy metrics (where they win) but not spatial distribution metrics (where they lose). The explanation is sound, though the performance gap persists.
- Scalability (bqeD): The authors showed that the number of angle tokens scales linearly with atoms ($O(N)$), but the quadratic cost of attention remains. The high pretraining cost (tens of A100 days) is acknowledged as a barrier, and promises to release checkpoints.

**Reviewer Scores:**

- Reviewer bqeD (Original: 4): Predicted: 6. The rebuttal provided several new experiments, e.g., the specific sensitivity data and geometric validity statistics this reviewer requested to assess soundness. The memory consumption is exploding with exponent=2, confirming the computational barrier concern.
- Reviewer nYFx (Original: 4): Predicted: 6 . The reorganization of the paper and the rationale for the QM9 trade-offs address the reviewer's critique regarding the analysis of results.
- Reviewer P24Z (Original: 6): Predicted: 6 . The review was short, and the terminology clarification resolves their main comment; likely to remain positive.
- Reviewer uB9E (Original: 6): Actual: 8. The reviewer explicitly raised their score in the forum after the efficiency data was added.

---

### Decision · Program_Chairs · 2026-01-26

Accept (Poster)